# Space variability impacts on hydrological responses of Nature-Based Solutions and the resulting uncertainty: a case study of Guyancourt (France)

Yangzi Qiu[1], Igor da Silva Rocha Paz[2], Feihu Chen[3], Pierre-Antoine Versini[1], Daniel Schertzer[1], Ioulia Tchiguirinskaia[1]

[1]Hydrology Meteorology & Complexity, École des Ponts ParisTech, Champs-sur-Marne,77455, France
[2]Instituto Militar de Engenharia, Rio de Janeiro, 22290-270, Brazil
[3]School of Architecture, Hunan University, Changsha, 410082, China

*Correspondence to*: Yangzi Qiu (yangzi.qiu@enpc.fr)

**Abstract.** During the last decades, the urban hydrological cycle has been strongly modified by the built environment, resulting in fast runoff and increasing the risk of waterlogging. Nature-Based Solutions (NBS), which apply green infrastructures, have been more and more widely considered as a sustainable approach for urban stormwater management. However, the assessment of NBS performance still requires further modelling development because of the hydrological modelling results strongly depend on the representation of the multiscale space variability of both the rainfall and the NBS distributions. Indeed, we initially argue this issue with the help of the multifractal intersection theorem. To illustrate the importance of this question, the spatial heterogeneous distributions of two series of NBS scenarios (porous pavement, rain garden, green roof, and combined) are quantified with the help of their fractal dimension. We point out consequences of their estimates. Then, a fully-distributed and physically-based hydrological model (Multi-Hydro) was applied to consider the studied catchment and these NBS scenarios with a spatial resolution of 10 m. Two approaches for processing the rainfall data were considered for three rainfall events: gridded and catchment-averaged. These simulations show that the impact of spatial variability of rainfall on the uncertainty of peak flow of NBS scenarios ranges from about 8 % to 18 %, which is more significant than those of the total runoff volume. In addition, the spatial variability of the rainfall intensity at the largest rainfall peak responds almost linearly to the uncertainty of the peak flow of NBS scenarios. However, the hydrological responses of NBS scenarios are less affected by the spatial distribution of NBS. Finally, the intersection of the spatial variability of rainfall and the spatial arrangement of NBS produces a somewhat significant effect on the peak flow of green roof scenarios and the total runoff volume of combined scenarios.

# 1 Introduction

The increased risk of flooding from urban storms appears to be closely linked to two key factors: rapid urbanization and climate change (Lovejoy and Schertzer, 2013). Adapting to climate change and mitigating urban flooding constitute now significant societal challenges (Loukas et al., 2010; Miller and Hutchins, 2017). Impervious surfaces directly connected to grey infrastructures result in a rapid transfer of rainfall into runoff, which greatly increases the risk of flooding, especially in urban watersheds (Fry and Maxwell, 2017; Ercolani et al., 2018). Expanding and upgrading the capacity of existing drainage systems has proven to be costly and unsustainable, which is challenging to realize in highly urbanized cites (Qin et al., 2013).

Increasing urban resilience to reduce the risk of urban flooding has been emphasized in many countries (e.g., Kelman, 2015). Nature-Based Solutions (NBS) refer to a sustainable strategy, capable to reduce the influences of human activities on the natural environment, especially efficient for stormwater management (European Commission, 2015; Cohen-Shacham et al., 2016). To some extent, the NBS concept builds on and supports similar widely used concepts (Bozovic et al., 2017), like the Low Impact Development (LID), or Blue Green Infrastructure (BGI), as well as some more local ones, like the Water Sensitive Urban Design (WSUD) from Australia (Morison and Brown, 2011) or 'Sponge city' proposed recently in China (Chan et al., 2018). Regarding stormwater management, NBS suggests using a suite of small-scale controlled measures. This often includes bio-retention swale, porous pavement, green roof, rain garden, and rain barrel, because these infrastructures are able to conserve or recover the natural environment of a region (Newcomer et al., 2014).

The hydrological performances of such NBS have been approached in terms of the reduction of total runoff volume and peak flow at the urban catchment scale (Zahmatkesh et al., 2015; Ahiablame and Shakya, 2016; Bloorchian et al., 2016). Generally, the results of a large number of studies are based on lumped or semi-distributed models (Ahiablame et al., 2013; Liu et al., 2015; Massoudieh et al., 2017; Guo et al., 2019). Indeed, as underlined by Fry and Maxwell (2017), and Her et al. (2017), fully-distributed models are rarely used (Versini et al., 2016; Hu et al., 2017; Versini et al., 2018). While there is a general consensus that these models should better assess the hydrological performances of NBS implemented at smaller scales, the deployment of the fully distributed models has been stuck for some time by three main factors: *(i)* availability of reliable high resolution forcing, *(ii)* complex interactions between the processes, and *(iii)* reliable parameterisation process (e.g., Imhoff et al., 2020). As a consequence, the semi-distributed Storm Water Management Model (SWMM) remains the one that is most frequently used to investigate the impact of NBS on urban runoff and water quality (Sun et al., 2014; Jia et al., 2015; Palla and Gnecco, 2015; Cipolla et al., 2016; Kwak et al., 2016). Nevertheless, Rossman et al. (2010) demonstrated that SWMM has some serious limitations for reflecting the heterogeneity of urban watersheds, which in turn presents some difficulties to sustainably replicate hydrological responses to various urban land uses. In particular, the study of Burszta-Adamiak and Mrowiec, (2013) confirmed that SWMM is not really explicit for presenting the hydrological responses of catchments with only the help of the percentage of pervious and impervious land covers. These gaps imply strong limitations to the results obtained with the help of lumped and the semi-distributed models. Thus, to make the modelling results more accurate and credible, there is a strong need to use fully-distributed and physically-based models.

At the same time, due to the long-standing challenge of the availability of reliable and high-resolution spatio-temporal precipitation measurements in urban areas, some studies have been devoted to assess the performance of NBS under the simplifying assumption of an uniform rainfall, hence the impact of spatial rainfall variability in the heterogeneous urban context has not been considered (Holman-Dodds et al., 2003; Gilroy and McCuen, 2009; Qin et al., 2013; Versini et al., 2018; Zhu et al., 2019; Guo et al., 2019). A strong impact of the temporal variability of precipitation on the response of the watershed is generally well recognised (Schertzer et al., 2010; Ochoa-Rodriguez et al., 2015; Gires et al., 2015). Qin et al. (2013) also investigated the performance of some NBS, such as swales, porous pavements and green roofs, as a function of peak precipitation intensity. Whereas the temporal variability of precipitation, even intuitively, forces the dynamics of the retention capacity of the NBS, the impact of the spatial variability of precipitation in the heterogeneous urban context has not yet been studied in its full extent. However, the hydrological responses of NBS (model outputs) can largely depend on: (i) the highly spatially variable rainfall fields, (ii) the spatial distribution of the NBS, and (iii) their intersection. Indeed, the rainfall and the NBS represent two heterogeneous fields that do not coincide, which implies that the overall performances of NBS scenarios simulated with uniform rainfall or lumped/semi-distributed model may not be entirely convincing. Therefore, the mentioned impacts remain to be investigated, especially for higher spatial model resolutions, using spatio-temporal rainfall fields with a fully-distributed model, allowing heterogeneous NBS scenarios.

In this respect, the main goal of this study is to investigate the uncertainty of hydrological responses in various NBS scenarios resulting from the spatial variability of rainfall and the heterogeneous distribution of NBS at the urban catchment scale, and thus not those associated to the model structure, hypothesis or parameterization for instance. A fully-distributed and physically-based hydrological model Multi-Hydro (Giangola-Murzyn, 2013; Ichiba et al., 2018) is applied on a semi-urban catchment of 5.2 km² in the city of Guyancourt city (France) at the scale of 10 m. The particularity of Multi-Hydro is its scalability, which makes it possible to replace the traditional parameter calibration by the process of rapid optimization of the spatio-temporal resolution of the model to ensure its best performance for the case study, based on the overall scaling of available data (Ichiba et al., 2018). Multi-Hydro is therefore well suited to achieve the desired objective of this study. Two different rainfall processing approaches (gridded and catchment-averaged) from three typical rainfall events of the Paris region are used as meteorological inputs: (i) based on the gridded approach, the data are retrieved from the X-band polarimetric radar of École des Ponts ParisTech (ENPC), characterized by high spatial and temporal resolutions, which are called distributed rainfall data; and (ii) the corresponding uniform rainfall data are obtained by catchment-averaged of the distributed rainfall data at each time step. The spatial heterogeneity of NBS is grasped by different landuse scenarios, characterized with the help of an across-scale indicator, the fractal dimension. This variability and resulting uncertainties in hydrological responses of the catchment are quantified by considering the peak flow and the total runoff volume in the drainage conduits. It is important to mention here that a precise quantitative evaluation of NBS performances, e.g., peak discharge reduction, total runoff volume reduction, or both, is not the goal of the present study. The authors aim first to deepen the knowledge on the impact of spatial variability of the rainfall on hydrological responses of several NBS scenarios, and that in turn helps to clarify whether the Nature-Based Solutions could be randomly implemented in semi-urban catchments or not.

The organization of this paper is as follows. The next section presents the study area, the rainfall data and the Multi-Hydro model. The details of the NBS scenarios, the framework of modelling experiments and the model validation are described in Section 3. Then, the obtained results are discussed in Section 4. Finally, the main conclusions are summarized in Section 5.

## 2 Study context, data and methods

### 2.1 The choice of the case study

This study is conducted on a semi-urban catchment, a part of the city of Guyancourt (France), located on the Saclay Plateau in the southwest suburb of Paris (Fig. 1). The available raw 25-m resolution Digital Elevation Model (DEM) obtained from the French National Institute of Forest and Geographic Information (IGN), which presents the whole catchment, is relatively flat (see the left side of Fig. 1). The altitude in the North is slightly higher than that of the South. The highest altitude in the whole catchment reaches 175.1 m, while the lowest one of 143.39 m corresponds to the location of the storage basin (i.e., the outlet

of the catchment: Etang des Roussières). The most recent statistical report of Météo-France (2020) indicates that the area is characterized by an oceanic climate with an average annual temperature of about 10.7°C and total annual precipitation around 695 mm. In this context, the Guyancourt catchment is an interesting and appropriate case study for several reasons.

*Firstly,* Guyancourt is one of the sub-catchments in the upstream of the 34.6 km long Bièvre River, which flows through several increasingly urbanized areas and joins the Seine River in Paris. Bièvre River is well-known by its drastic contribution

to the historical 1910 flood in Paris and still easily generates flash floods during the heavy rainfall events (e.g., two severe floods occurred in 1973 and 1982). Therefore, the case of Guyancourt has a reference significance for the Paris region.

*Secondly*, the Guyancourt city is expected to become a part of the "French Silicon Valley", which currently undergoes a rapid urbanization process over its total area of around 5.2 km$^2$, with a population of about 30,000 (INSEE 2020). Based on the data from IGN, the current land use of the study area consists of seven main types, including road, parking, building, gully, forest,

grass, and water. In total, these seven land use types cover 9.6 %, 10.6 %, 15.5 %, 1.9 %, 28.8 %, 32.7 %, and 0.9 % of the total area, respectively, as shown on the left side of Fig. 2. Currently, the pervious surface accounts for 62.4 % of the total area, and the corresponding impervious surface is 37.6%.

The local authority, the agglomeration community of Saint-Quentin-en-Yvelines ("La communauté d'agglomération de Saint-Quentin-en-Yveline"), manages the urban drainage system of the catchment and provided some related data (right side of Fig.

2). The total length of the drainage system is about 76 km and consists of 4,474 nodes and 4,534 conduits. Overall, the drainage system was designed with a capacity characterized by a return period ranging from 2 to 10 years. The diameters of conduits range between 0.1 m to 1.6 m, 70 % of them between 0.3 to 0.5 m (marked with a yellow line on the right side of Fig. 2). The conduits with a diameter ranging between 0.9 to 1.6 m (marked with a purple line on the right side of Fig. 2) are the primary conduits, which converge the flow to the storage basin and the outlet. The rainfall amount corresponding to the mentioned

return periods (from 2 to 10 years) depends on the considered duration (usually equal to the concentration time). So, this duration value depends on the location of pipes in the catchment and its upstream area. Here are the corresponding values for

different durations that can be found on the studied watershed (by using the Montana coefficients): *(i)* Duration 5 minutes: 187 mm/h for T = 10 years and 125 mm/h for T = 2 years; *(ii)* Duration 30 minutes: 50 mm/h for T = 10 years and 31 mm/h for T = 2 years; *(iii)* Duration 1 hour: 30 mm/h for T = 10 years and 18 mm/h for T = 2 years; *(iv)* Duration 2 hours: 20mm/h for T=10 years and 13 mm/h for T=2 years.

Regarding the properties of the three selected rainfall events (Table 1), the drainage system seems to have the capacity to drain the rainfall intensities on these durations. Nevertheless, we do not have any information about the exact duration range that was considered for the design (durations smaller than 30 minutes are usually not considered).

However, due to climate change, a clear tendency towards a growing number of shorter duration, but higher intensity rainfall events is perceived for this region (Hoang et al., 2010), causing in recent years a large amount of fast surface runoff and higher peak flow rates. The existing stormwater drainage system may not be able to sustain the future modifications of the watershed, and some low-lying areas in the catchment could suffer more easily from waterlogging, even during moderate rainfalls. As displays Fig. 1, some vulnerable areas and buildings subject to a risk of waterlogging were defined in the Guyancourt catchment by using the ModelBuilder of ArcGIS software (a geoprocessing model for identifying landscape sinks [https://learn.arcgis.com/en/]). This geoprocessing model is based on a sequential chain of GIS analysis tools. By exploring the Digital Elevation Model (DEM) of the Guyancourt catchment with the ArcGIS Desktop hydrology tools (https://desktop.arcgis.com), we first identify the landscape sinks. On this figure, the blue spots represent the low-lying areas with a total area of 0.6 km$^2$ that can be easily flooded by stormwater (average rainfall depth of 53 mm). Then, the locations of the landscape sinks can be compared with the locations of existing buildings, and the buildings that are situated inside or adjacent to the landscape sinks are defined as the vulnerable buildings. Correspondingly, the yellow spots indicate these vulnerable buildings on the figure.

*Thirdly*, the local authority installed a gauge at the storage basin (outlet) to monitor water levels, which provided a measurement point of the Guyancourt catchment.

Overall, the relative complexity of the catchment makes it a typical "case study" for analysing some of the uncertainties related to hydrological responses of NBS scenarios, aiming to help the local authorities to find more reasonable and ecological alternatives for future urban planning.

## 2.2 Rainfall data

In this study, one of the purposes is to assess the impact of spatial variability of rainfall on the hydrological responses of some NBS scenarios. Hence, two approaches for processing rainfall data were used to prepare the meteorological inputs: gridded (distributed) and catchment-averaged (uniform). Based on the gridded approach, the distributed rainfall data were retrieved from the polarimetric X-band radar, located in ENPC, Champs-sur-Marne (East of Paris, France). The distance between the X-band radar and the catchment is around 45 km (see Fig. 1). The spatial and temporal resolutions of the X-band radar are 250 m and 3.4 min, respectively. Three relatively long rainfall events (EV1, EV2, and EV3) with different characteristics that occurred in 2015 were chosen for the study (see Table 1 for more details).

Figure 3 (top) shows the maps of rainfall intensity (per radar pixel) at the largest rainfall peak for these three events. The maximum rainfall intensity per pixel is 41.2, 29.1, and 55.6 mm h$^{-1}$, respectively. To establish a link with classical approaches (e.g., Hamidi et al., 2018), the standard deviation (*SD*) was used to quantify the variability of the rainfall fields. As presented in Table.1, the *SD* of the rainfall intensity at the largest rainfall peak of the three rainfall events is 4.31, 6.11, and 5.75 mm h$^{-1}$, respectively. This illustrates that while the strongest rainfall intensity was observed during the EV3, the highest variability

of rainfall intensity occurred in the EV2. Figure 3 (middle) presents the total (cumulative) rainfall depth (per radar pixel) for the three rainfall events. The maximum cumulative rainfall per pixel is 36.9, 14.1, and 25.4 mm, respectively. The *SD* of the total rainfall depth of the three rainfall events is 1.21, 0.82, and 1.35 mm, respectively. This demonstrates that the spatial distributions of cumulative rainfall are much less variable compared to those of the rainfall intensity at the peak, with the highest variability computed for the EV3.

The three uniform rainfall events (EV1U, EV2U and EV3U) were constructed by spatial averaging over the whole catchment of originally distributed rainfall fields at each time step. Figure 3 (bottom) presents the time evolution of the corresponding rainfall rates and cumulative rainfall depths. Each of these events is sufficiently long to contain several rainfall peaks and dry periods. For EV1U, the highest rainfall intensity reaches 20 mm h$^{-1}$, and the total rainfall accumulates (around 31.5 mm) fast between the first and the third rainfall periods (approximately 24 h). The maximum rainfall intensity of the EV2U and EV3U

is 9 mm h$^{-1}$ and 36.4 mm h$^{-1}$, and the total rainfall amounts about 12 mm and 20 mm, respectively. Although the largest rainfall peak of the EV3U is 36.4 mm h$^{-1}$, it lasted only for 3 min, just sufficient to contribute about 10 % to the total rainfall depths. Overall, this initial analysis suggests that in spite of some similar characteristics, the selected events cover a truly wide spectrum of rainfall space-time variability. However, to deepen the knowledge of intersection effects between the spatial variability of rainfall and spatial distribution of NBS, we also considered four synthetic rainfall events (EV4 – EV7). All these

events are based on the EV3U, by selecting the 2 hours period with the highest rainfall peak around 35 mm h$^{-1}$ (catchment-averaged), as illustrated on Fig. 4a. However, during the 3 min that lasted the largest rainfall peak of the EV3U, a new space distribution and/or intensity of the rainfall was imposed for each synthetic rainfall event. As shown in Fig. 4b, the catchment-averaged maximum rainfall peak is about 37 mm h$^{-1}$ for the EV4, and the corresponding catchment-averaged cumulative rainfall is about 4 mm. During these 3 minutes, the rainfall was binary re-distributed in space (Fig. 4c), with the maximum

intensity around 55 mm h$^{-1}$. For the remaining synthetic rain events, this binary distribution was modified as follows (see Fig. 4d-f): the same maximum intensity of 55 mm h$^{-1}$ and zero rainfall elsewhere (EV5), the maximum intensity of 17 mm h$^{-1}$ and zero rainfall elsewhere (EV6), and the maximum intensity of 55 mm h$^{-1}$ has been replaced by zero rainfall (EV7).

### 2.3 Multi-Hydro model

The Multi-Hydro model is a fully-distributed and physically-based hydrological model, which has been developed by
190 HM&Co/ENPC (El Tabach et al., 2009; Giangola-Murzyn, 2013; Ichiba, 2016; Ichiba et al., 2018). It has been successfully implemented and validated in several catchments (e.g., Versini et al., 2016; Ichiba et al., 2017; Gires et al., 2017; Gires et al.,

2018; Alves de Souza et al., 2018; Versini et al., 2018; Paz et al., 2019). In this study, it is used for assessing hydrological responses of the NBS scenarios at the urban catchment scale.

Multi-Hydro constitutes the interactive core among the four open-source modules (rainfall, surface, groundwater, drainage) that represent essential elements of the hydrological cycle in urban environment.

The rainfall module (MHRC) can treat different kinds of rainfall data (from radar or rain gauge). In order to adapt the rainfall inputs for Multi-Hydro, the intersection between the pixels of the model (with a 10 m spatial resolution in this study) and the pixels of the X-band radar data (with a 250 m spatial resolution) were performed by the QGIS interface using the following equation (Paz et al., 2018):

$$R_{i_M,j_M} = \frac{\sum_{i_X,j_X} R_{i_X,j_X} |A_{i_M,j_M} \cap A_{i_X,j_X}|}{|A_M|} \tag{1}$$

where $R_{i_M,j_M}$ is the rainfall rate computed on the model pixel $A_{i_M,j_M}$ of coordinates $(i_M, j_M)$; $R_{i_X,j_X}$ is the rainfall rate measured by the X-band radar on its pixel $A_{i_X,j_X}$ of coordinates $(i_X, j_X)$; $|S|$ denotes the surface of any pixel $S$, in particular $|A_M|$ is the surface of the model pixel (it does not depend on the coordinates, but only by the model resolution).

The surface module (MHSC) of Multi-Hydro uses the code of the Two-Dimensional Runoff Erosion and Export (TREX) model that computes the interception, storage and infiltration occurring at each pixel in terms of the properties of each land use (Velleux et al., 2008). The infiltration process of the surface module is governed by simplification of Green and Ampt equation (see p. 4 of the TREX user manual). The diffusive wave approximation of the Saint-Venant equations is used for calculating the overland flow, following the conservation of mass and momentum equations.

The groundwater recharge and solute transport (Riva et al., 2006; Mooers et al., 2018) are the other significant aspects of the hydrological cycle. The groundwater module (MHGC) is based on the Variably Saturated and 2-Dimensional Transport (VS2DT) model developed by the U.S. Geological Survey. This module can be used to simulate variably saturated transient water flow and solute transport in one or two dimensions (Lappala et al., 1987; Healy, 1990). The drainage module (MHDC) in Multi-Hydro uses the code of 1D SWMM model (James et al., 2010) to simulate the sewer network. This model represents the flow computed by 1D Saint-Venant equations in conduits and nodes.

In this study, we used the Multi-Hydro interaction between the surface module and the drainage module to focus on the rainfall-runoff modelling of NBS scenarios. In urban areas, groundwater can produce infiltration into the drainage pipes due to cracks in the structure (see Lucas and Sample, 2015 for an example). The absence of long recession limb on the hydrographs indicates there is no such problems on the studied watershed. Groundwater can also eventually contribute to surface flooding when it is saturated. Such phenomenon did not occur on the studied area due to its pedology and the considered (not extreme) rainfall events. For these reasons, groundwater (as evapotranspiration) has not been considered in this study. That has been focused on the fast response of the watershed at the rainfall event scale.

The high spatial resolution of Multi-Hydro allows an easy implementation of small-scale controlled measures, like the rain garden, green roof, bio-retention swale, porous pavement, and rainwater tank, by locally modifying the land use parameters to link the size and shape of the corresponding NBS infrastructures, with their infiltration and storage capacities.

## 3 Numerical investigation of the NBS scenarios

### 3.1 Multi-Hydro implementation in the Guyancourt catchment

Based on the fully distributed character of Multi-Hydro, users can choose a specific spatial resolution. In this study, Multi-Hydro was implemented with a 10 m spatial resolution (the grid system creates square grids with a cell size of 10 m), and a temporal loop of 3 min. The 10 m resolution was performed because it sufficiently represents the heterogeneity of the catchment, and also for saving the computation time.

In analogy with Eq. 1, the rainfall input for Multi-Hydro has been also time interpolated from the X-band radar measurements, as follows:

$$R_m(j) = \frac{\sum_i R_r(i)|\Delta_m(j) \cap \Delta_r(i)|}{\delta_m} \tag{2}$$

where $R_m(j)$ is the rainfall rate during the j-th time interval $\Delta_m(j)$ of the model, $R_r(i)$ is the rainfall rate during the i-th time interval $\Delta_r(i)$ of the X-band radar. $|\Delta|$ denotes the length of any interval $\Delta$ and $\delta_m = |\Delta_m|$ is the length of any time interval of the model. Note that while the duration of the time loop to generate the model outputs is 3 min (to keep it comparable with the X-band radar time interval), $\delta_m = 1$ minute for the rain input to Multi-Hydro.

The implementation of Multi-Hydro in a new catchment starts with the conversion of the original GIS data (e.g., land use, topography) into the standard rasterised format with the desired resolution by using the MH-AssimTool (Richard et al., 2013), a supplementary GIS-based module for generating the input data for Multi-Hydro. During this process, a unique land use class was assigned to each pixel, specifying its hydrological and physical properties. In order to attribute a unique land use class to each pixel, the following priority order was used in this study: gully, road, parking, house, forest, grass, and water surface. For this study, all the standard model parameters related to the land use classification were selected from the Multi-Hydro manual (Giangola-Murzyn et al., 2014). The most important parameters are Manning's coefficient (no unit), hydraulic conductivity (m s$^{-1}$) and interception (mm), as they are shown in Table 2. As already indicated, the Multi-Hydro does not use the traditional calibration of these parameters. If their most common values are always used, the reliable heterogeneity of the watershed for each case study is obtained by a rapid optimization of the spatio-temporal resolution of the model, with possibly refined classes of the land use and their orders (Ichiba et al., 2018).

Since the gully is actually the only land use class able to connect the surface module and the drainage module, it has the highest priority (i.e., if a raster pixel contains gully and the other land use classes, the whole pixel will be considered as gully). Generally, this order considers the impervious land use classes have higher priority than the permeable land use classes, which result in an overestimation of impervious land uses (see Ichiba et al., 2017, for an alternative approach). After the rasterization process, the impervious land uses occupied 54 % of the Guyancourt catchment (Fig. 5). Besides the land use, the elevation was also assigned to each pixel of the model. For this purpose, the interpolation method was used to downscale the raw DEM data from 25 m to 10 m (DEM25-10) to incorporate with the model resolution. More precisely, each pixel was first subdivided

into 25 equal sub-pixels as a proxy of the 5 m resolution, then the elevation data were up-scaled 4 by 4 pixels to produce the 10 m interpolation of the original elevation.

While the 25 m resolution DEM may seem too coarse to use for an urban area, it did not limit the study in any way because the catchment is relatively flat. To test this, we up-scaled the raw 5 m DEM data to adapt them to the model resolution (DEM5-10). Table 3 presents the results of the statistical analysis of DEM25-10 and DEM5-10, which are so similar that the difference could not impact the results. For instance, the Root Mean Square Error is about 0.26, and the correlation coefficient is around 0.99. Besides, the ensemble of the data actually available for the Guyancourt watershed would need to be more detailed to make it worth going to a higher resolution of the model.

As the most considered NBS correspond to more specific land uses, they are characterised with different retention capacities, the related parameters are based on the literatures (Dussaillant et al., 2004; Kuang et al., 2011; Park et al., 2014). To be more specific, the rain gardens (RG) characterised with the depression depth of 0.3 m. Thus, the storage capacity of RG is about 300 L $m^{-2}$. For the porous pavements (PPs), the thickness of pavements is 0.21 m (i.e., pavement (0.08 m), bedding material (0.03 m) and base material (0.1 m)). The porosity of pavement, bedding material, and base material is 5.4 %, 28.29 % and 22.66 %, respectively. This indicates that the storage capacity of PP is approximately 74 L $m^{-2}$ in this study. For these two NBS measures, a simple procedure represents both infiltration and storage processes has been carried out. For each time step, if the rainfall rate lower than infiltration rate of porous pavement/rain garden, the water is stored. If not, then the ponding occurs.

Green roof is a special NBS measure that can be simulated by a specific module in Multi-Hydro (Versini et al., 2016). Accordingly, five physically-based parameters are defined for the green roof. They are based on the experimental site of Cerema (Ile-de-France) where several green roof configurations were monitored (see Versini et al., 2016). In detail, the chosen configuration is the following: substrate thinness of 0.03 m and characterized by a porosity of 39.5%, an initial moisture condition of 10 %, a field capacity of 0.3, and a hydraulic conductivity of 1.2 m $h^{-1}$.

## 3.2 Simulation scenarios

For achieving the purpose of the study, a series of NBS scenarios were created and simulated for both rainfall inputs (described in Sect. 2.2). The baseline scenario is considered as the current configuration of the Guyancourt catchment, without implementing any NBS (Fig. 2 left). The baseline scenario will be used later on for the model validation.

The first set of NBS scenarios includes porous pavement (PP1), rain garden (RG1), green roof (GR1), and their combined scenario (Combined1). They are applied to assess the impacts of the spatial variability of rainfall on the hydrological responses of NBS scenarios. For each scenario, the corresponding NBS are implemented heterogeneously over the catchment, while respecting the local catchment conditions and stormwater management requirements. For instance, with the help of the detailed land use GIS data, we initially selected all the buildings having flat roofs, then these impervious roofs were converted into green roofs for the GR1 scenarios by adapting the land use data.

The second set of NBS scenarios (PP2, RG2, GR2, and Combined2) was proposed with a different arrangement to assess the potential effects of a heterogeneous implementation of NBS at the urban catchment scale. For each pair of scenarios with a

given type of NBSs, their implementation occupies the same percentage of the space over the whole catchment but differs significantly in terms of spatial distributions of the considered asset. Considering now the roofs with certain slopes (≤ 15°), they can be also used to implement green roofs (Stanic et al., 2018). The impervious roofs that satisfied this condition were converted into small and light green roofs and used for the GR2 scenario. While the two scenarios (GR1 and GR2) occupy the same percentage of the whole catchment, their density is different, simply because of the difference of original densities of the buildings. The designing process for other NBS scenarios follows a somewhat similar logic. All details concerning the scenarios of the NBS implementations, including a detailed description of each NBS and the percentage of the space required for its implementation at 10 m resolution, are presented in Table 4, while the maps of the resulting land use are illustrated on Fig. 6.

Table 4 provides also the estimates of the scale independent indicator, discussed in detail in the following sections, called the fractal dimension. To get it intuitively, this indicator for the two combined scenarios (Combined1 and Combined2) is close to 2 over the range of large scales of the 2-dimensional space. This indicates that NBS are rather homogeneously implemented over the whole catchment. However, it is important to note that, in spite of initially identical percentage that has been used to characterise the implementation of the NBS pairs over the catchment at a 10 m scale, the resulting fractal dimension could be quite different. It is simply because the percentage of the space required for the NBS implementation remains a scale dependent quantity, i.e., it depends on the resolution of the model, while the fractal dimension quantifies the propagation of the spatial heterogeneity for each of NBS scenarios, from the smallest scale to the outer scale of the catchment. This propagation remains scenario dependent only and hence a subject to its optimisation.

### 3.2.1 Fractal dimension of NBS scenarios

To quantify the multi-scale space heterogeneity of NBS in each NBS scenario, we applied the concept of fractal dimension ($D_F$), which was initially introduced to describe the scale invariance of some irregular geometric objects (Mandelbrot, 1983). Namely, a similar structure can be observed in any scale. $D_F$ has been often used in catchment hydrology (e.g., Schertzer and Lovejoy, 1984; Schertzer and Lovejoy, 1987; Schertzer and Lovejoy, 1991; Lavallée et al., 1993; Gires et al., 2013; Gires et al., 2016; Ichiba et al., 2017; Paz et al., 2020; Versini et al., 2020). In this study, a standard box-counting technique was applied to estimate the $D_F$ of each NBS scenario (Hentschel and Procaccia, 1983; Lovejoy et al., 1987). The $D_F$ of a geometrical set $A$ (here represented by the non-overlapping pixels of NBS embedded in a 2-dimentional space) is obtained with the following power-law:

$$N_{\lambda,A} \approx \lambda^{D_F} \qquad (3)$$

where $N_{\lambda,A}$ is the number of non-empty (containing NBS) pixels to cover the set $A$, at the resolution $\lambda$, which is defined as the ratio between the outer scale $L$ and the observation scale $l$ ($\lambda = \frac{L}{l}$). The symbol $\approx$ means an asymptotic relation (i.e. for large resolution $\lambda$ and possibly up to a proportionality prefactor).

Based on Eq. (3), we count the number of pixels containing at least one NBS, by starting with the smallest pixel size ($l = 10$ m in this study), then continuously increasing the pixel size by simply merging the 4 adjoined pixels. This procedure is repeated until reaching the largest pixel size ($L$). Thus, $N_{\lambda,A}$ is counted at different resolutions, and the results are plotted in the log-log plot (see Fig. 7). Corresponding to Eq. (3), the fractal dimension $D_F$ of each NBS scenario is defined as follows:

$$D_F \approx \frac{ln(N_{\lambda,A})}{ln\,\lambda} \tag{4}$$

Here, for each scenario, a square area of 128 x 128 pixels was extracted from the catchment to make the fractal analysis (see the example of the PP1 scenario in Fig. 6). In order to avoid the "no data" areas, which would bias the fractal dimension estimate, the selected square area is the greatest possible size characterized by a multiple of two in the studied catchment.

As shown in Fig. 7, all the NBS scenarios are presented with two scaling behaviour regimes, with a scale break roughly at 80 m. For each regime, the scaling is robust, with linear regression coefficients ($R^2$) around 0.99. For the first regime

corresponding to the small-scale range (10 m – 80 m) that related to the asset implementation level, the dimension $D_F$ is around 1 for most of NBS scenarios. It is in contrast with the second regime, the large-scale range (from 80 m to 1280 m) that exhibits a scaling behaviour with a $D_F$ ranging from about 1.75 to 1.98. We also applied fractal analysis on the impervious surface of the baseline scenario in the same selected area, and we also found the same scale break at 80 m (the $D_F$ of the baseline scenario in each regime are presented in Fig. 7). Therefore, it rather confirms that the spatial distribution of NBS is strongly constrained

by the urbanisation level of the catchment.

### 3.2.2 Multifractal intersection theorem

We would like now to illustrate and emphasise why it is so much indispensable to take into account the multiscale space variability of both the rainfall and the NBS distribution. For instance, both "hot spots" (extremes) of the rainfall and NBS are scarce and therefore could rarely coincide, i.e., rainfall spikes may fall more often elsewhere than on NBS. Similar questions

can occur for less extreme events. The effective NBS performance could be therefore biased with respect to their potential performance due to this problem of intersection between rainfall intensity and NBS. It reminds us of the so-called multifractal intersection theorem applied to the intersection of a rainfall with extreme space variability and a rain gauge network that provides quantitative estimates of this intersection (Tchiguirinskaia et al., 2004). Figure 8, adapted from this paper, schematically represents the intersection at a given time of a (multifractal) rainfall, displaying quite variable pixel intensities

ranging from light blue to dark brown (e.g., from 1 to 100 mm h$^{-1}$), with a heterogeneous rain gauge network (light brown pixels). The resulting measured rainfall field $M$ is simply the product of the rainfall intensities $R$ by the gauge characteristic function $N$ (=1 if there is a gauge in this pixel, 0 otherwise). The intersection theorem states that for fractal objects, like for the usual (Euclidean) geometric ones, the codimension – i.e. the complement $c_M = d - D_M$ of the dimension $D_M$ to the embedding space dimension $d$ – of the measured field above a given intensity threshold is the sum of the codimensions of the network

($c_N = d - D_N$) and of the "real" field ($c_R = d - D_R$) above the same intensity threshold:

$$c_M = c_N + c_R \Leftrightarrow D_M = D_N + D_R - d \tag{5}$$

For instance, the intersection in a plane ($d = 2$) of two straight lines ($D_N = D_R = 1$; $c_N = c_R = 1$) corresponds to a point ($c_M = 2$, $D_M = 0$). Of particular interest is the case where the intersection is so scarce that its codimension $c_M$ is larger than the embedding dimension $d$, i.e. has a negative dimension $D_M$ (Schertzer and Lovejoy, 1987). Due to Eq. (5), the codimension of the network $c_N$ is thus the critical dimension of the (real) field under which the rainfall intensity is rarely measured by the network:

$$c_M > \mathrm{d} \Leftrightarrow D_M < 0 \Leftrightarrow D_R < c_N = d - D_N \tag{6}$$

More precisely, the smaller $D_R$ is with respect to $c_N$, the rarer the real field $R$ is measured. Let us mention that Paz et al., (2020) used this intersection theorem to determine when the adjustment of radar data by a rain gauge network becomes misleading instead of improving the data.

The assessment of the performance of an NBS network cannot be reduced to the binary question of presence or not of an NBS, like done for a rain gauge of a network. However, we can immediately state they will be more and more ineffective for rainfall intensity whose fractal dimension is more and more below the codimension $c_N$ of the network. This is already an important information that can be used to design NBS and their networks. This also explains why we estimated in the previous subsection the fractal (co-) dimension of the NBS network, as well as to compare in section 4.3 simulations resulting from spatially uniform rainfalls ($D_R = d$, $c_R = 0$) and spatially heterogeneous rainfalls ($D_R < d$, $c_R > 0$).

## 3.3 Modelling experiments

The overall target of the study is to investigate whether the spatial variability of rainfall and the spatial arrangement of NBS have an impact on the hydrological responses of NBS scenarios at the urban catchment scale. For this purpose, three sets of modelling experiments were prepared, and two indicators ($PD_{Qp}$, percentage difference on peak flow; $PD_V$, percentage difference on total runoff volume) were used for quantifying the uncertainty associated to rainfall and NBS spatial distribution in the hydrological response of the catchment. Figure 9 presents the flow chart of the four sets of modelling experiments. In addition, the corresponding descriptions are presented as follows:

*The first set* is used to investigate the impact of spatial variability of rainfall on the hydrological responses of NBS scenarios. In this first set, we employed the following scenarios: baseline, PP1, RG1, GR1, and Combined1. These five scenarios were simulated under the distributed and uniform rainfall inputs. Then, we computed the ratio of peak flow (Eq. (7)), and the $PD_{Qp}$ and $PD_V$ indicators (Eq. (8) and Eq. (9)) for each scenario under two different kinds of rainfall inputs.

*The second set* is used to analyse the impact of the spatial distribution of NBS on the hydrological responses of the NBS scenarios. In this experiment, we compared the two groups of NBS scenarios mentioned in the Section 3.2 (GR1 vs GR2 for instance). The eight scenarios were simulated only with the uniform rainfall in order to avoid the impact of spatial variability of rainfall and to focus on the uncertainty associated with the spatial arrangement of NBS.

*The third set* is used to analyse the intersection impact of spatial variability of rainfall and the spatial distribution of NBS on the hydrological responses of the catchment. In this experiment, the eight mentioned NBS scenarios were simulated under the distributed and uniform rainfall, respectively. Then, the $PD_{Qp}$ and $PD_V$ of each NBS scenario were computed by comparing

the results obtained for the two different kinds of rainfall inputs (distributed and uniform). Then, we compared the difference of $PD_{Qp}$ and the difference of $PD_V$ between the NBS scenarios characterized by the same solutions/measures.

*The fourth set* is to verify the generality of the results obtained, we extended the study of hydrological responses to the intersection of the distributed rainfall and NBS by applying the synthetic rainfall events of EV4 – EV7 in two green roof scenarios (GR1 and GR2). The reason is the difference of $D_F$ between GR1 and GR2 is larger compare to the other NBS scenarios. Thus, the intersection effects can be more significant for these two scenarios. Here, the GR1 scenario was taken as the reference scenario, assuming that the extremes of rainfall (hot spots) only fall on the GRs of the GR1 scenario. With this respect, the rainfall was binary re-distributed in space during the 3 min that lasted at the largest rainfall peak of the EV3U, as illustrated on Fig. 4c-f. Namely, the 'hot spots' of the EV4 – EV6 are strictly intersected with the distributions of GRs in GR1, while the GR2 scenario is not. Contrary to EV4 – EV6, EV7 corresponds to the 'no rain' situation on GR1 during the same 3 min.

The peak flow ratio and the two indicators are especially calculated for the sum of four highlighted conduits connected to the catchment outlet (the right side of Fig. 2) with the following equations:

$$Ratio = \frac{Q_{p_1}}{Q_{p_2}} \tag{7}$$

$$PD_{Q_p}(\%) = \frac{|Q_{p_1} - Q_{P_2}|}{\frac{(Q_{p_1} + Q_{P_2})}{2}} \times 100 \tag{8}$$

$$PD_V(\%) = \frac{|V_1 - V_2|}{\frac{(V_1 + V_2)}{2}} \times 100 \tag{9}$$

where $Q_{p_1}$ and $Q_{P_2}$ refer to the peak flow of scenarios under the distributed rainfall and uniform rainfall respectively for the first and third modelling experiment. For the second experiment, they represent the peak flow of the first set of NBS scenarios and the second set of NBS scenarios, respectively. For the fourth set experiment, they represent the peak flow of the GR1 scenario and GR2 scenario, respectively. Correspondingly, for the first and third modelling experiments, $V_1$ and $V_2$ refer to the total runoff volume of scenarios under the distributed rainfall and uniform rainfall respectively. For the second modelling experiment, they represent the total runoff volume of the first set of NBS scenarios and the second set of NBS scenarios, respectively. For the fourth set experiment, they represent the total runoff volume of the GR1 scenario and GR2 scenario, respectively.

### 3.4 Model validation

Before the simulation of NBS scenarios, Multi-Hydro was validated with the water levels of the storage basin by applying the baseline scenario under the three distributed rainfall events. The simulations were then repeated with the three uniform rainfall events, respectively. The model performance was evaluated through two indicators: Nash-Sutcliffe Efficiency (*NSE*) and percentage error (*PE*). The Nash-Sutcliffe Efficiency (*NSE* ≤1) is an indicator generally used to verify the quality of the hydrological model simulation results, described as follows (Nash and Sutcliffe, 1970):

$$NSE(S_i, O_i) = 1 - \frac{\sum_{i=1}^{n}(O_i - S_i)^2}{\sum_{i=1}^{n}(O_i - \overline{O})^2} \tag{10}$$

where $S_i$ refers to simulated values, $O_i$ refers to observed values, and $\overline{O}$ represents the average of the observed values. The *NSE* closer to 1 indicates that the model is more reliable, whereas *NSE* closer to 0 indicates that the simulation does not better than that of the average observed value $\overline{O}$, which means the simulation performance is rather poor. If *NSE* is far less than 0, it means the simulation is even less performing than $\overline{O}$.

The percentage error (*PE*) represents the difference between observed values and simulation values, which reflects the reliability of the simulation values.

$$PE(S_i, O_i) = \frac{\sum_{i=1}^{n}|O_i - S_i|}{\sum_{i=1}^{n} O_i} \times 100\% \tag{11}$$

The values obtained for these two indicators are summarized in Table 5. They confirm that Multi-Hydro performs well for the case study area in the baseline scenario, suggesting that the model is reliable enough to study the impacts of spatial variability, either precipitation or/and NBS arrangements, on the hydrological responses under various NBS scenarios.

## 4 Results and discussion

### 4.1 Validation of the baseline scenario

Regarding the observed and simulated water levels in the baseline scenario, the model indeed performs well for the studied area. The *NSE* coefficients and the *PE* indicators validated Multi-Hydro's performance (see Table 5). For the three distributed rainfall events (Fig. 10), the *NSE* are larger than 0.9, and *PE* are lower than 5 %. For the uniform rainfall event of EV2U, the model represents the water levels with *NSE* equal to 0.95, and *PE* equal to 1.96 %: only a slight overestimation of the observed water levels is observed between hours 4 to 7. For the uniform rainfall of EV1U and EV3U, the temporal evolutions of simulated water levels slightly underestimate the observed ones, with *NSE* around 0.8, as well as *PE* around 7 %. Regarding the temporal evolutions of simulated water levels under the distributed rainfall of EV1 and EV3, they are more consistent with the observed ones. The reason is that the rainfall intensities of the distributed rainfall are generally higher than those of the uniform rainfall at the storage basin location. Namely, in the uniform rainfall events, the accumulated water levels in the storage basin are less than that of in distributed rainfall events. Overall, the distributed rainfall gives slightly better results, and the simulated water levels using uniform rainfall also match sufficiently well the observed ones to validate the Multi-Hydro implementation in the Guyancourt catchment.

Regarding the validation results, the scalability of Multi-Hydro allowed us to define the optimal resolution to finely reproduce the spatial heterogeneity of the watershed. Remember that this resolution is the ratio between the external scale of the watershed and the scale of the grid. The heterogeneity mentioned above propagates from the smallest scale to the largest, impacting the simulation results in any through the hierarchy of spatial scales of the watershed. It should be understood that the selected 10 m grid scale is not the smallest scale possible, but the optimal one to ensure a good balance between, for example, sufficient

heterogeneity and the required quantity of the data required, again in precision valuable and involved computing time involved. As discussed in Section 3.2, the spatial heterogeneity for each of the NBS scenarios evolves with the fractal dimension on two scale ranges: the asset implementation scales (10 m – 80 m) and the larger basin scales. Such an evolution remains fully compatible with the intrinsic scalability of Multi-Hydro, which makes it particularly suitable and sufficiently reliable to study the impacts of the spatial variability of hydrological responses in different NBS scenarios.

## 4.2 Impacts of spatial variability of rainfall

The impact of spatial variability of rainfall on the hydrological responses of each NBS scenario over the whole catchment was evaluated at the catchment scale in terms of the sum of flow in four conduits (highlighted on the right side of Fig. 2). These four conduits are chosen because they collect the runoff from the whole catchment and finally merge into the storage unit representing the outlet of the drainage system. To be more specific, the percentage difference on peak flow ($PD_{Qp}$) and

percentage difference on total runoff volume ($PD_V$) computed for the first set of modelling experiments (described in Sect. 3.3) are presented in the following section.

### 4.2.1 Baseline scenario

Before continuing, it is important to assess the 'baseline' scenario under both distributed and uniform rainfalls, by using the simulations already performed to validate the Multi-Hydro implementation in the Guyancourt catchment. As shown in the

hydrographs (Fig. 11), the higher peak flow was generated by the distributed rainfall in EV1 and EV2. Hence, the peak flow ratio computed by comparing distributed rainfall and uniform rainfall is larger than 1 (see the first column of Fig. 13a), but this ratio is around 0.9 in EV3. The reason is that during the largest rainfall peak of EV1 and EV2, the rainfall intensity of all radar pixels in distributed rainfall is higher than those of uniform rainfall. While in EV3, the rainfall intensity of around 30 % radar pixels in uniform rainfall is about 28 mm h$^{-1}$ higher than that of the distributed rainfall.

As shown in Fig. 13b, the $PD_{Qp}$ of baseline scenario in EV1, EV2 and EV3 is about 10 %, 17.6 %, and 11.6 %, respectively. According to the $SD$ of the rainfall intensity at the largest rainfall peak of each event (Table 1), the spatial variability of the rainfall intensity of EV2 is more pronounced than that of EV1 and EV3. Accordingly, the $PD_{Qp}$ of baseline scenario in EV2 is the highest. Regarding the total runoff volume (Fig. 13c), the $PD_V$ of the baseline scenario for the three rainfall events range from 1 % to 3.8 %. Contrary to the $PD_{Qp}$, the $PD_V$ of the baseline scenario is not correlated to the $SD$ of the total rainfall depth.

For the baseline scenario, it is noticed that the $PD_{Qp}$ is more pronounced than $PD_V$ for all rainfall events. These results can be explained by the fact that the spatial variability of rainfall intensity at the largest rainfall peak is strong in all three rainfall events, while the total rainfall volume for the distributed and uniform rainfall inputs is the same. This small $PD_V$ is influenced by the differences on the grid scale (storage capacity, infiltration, etc.), which are differently modelled when the input is uniform or non-uniform.

## 4.2.2 NBS scenarios

In comparison to Fig. 11, Fig. 12 presents the simulated flow of the first set of NBS scenarios under the three distributed rainfall and three uniform rainfall events. The results remain overall consistent with the results in the baseline scenario. Indeed, as shown in Fig. 13a, the peak flow ratios between distributed rainfall and uniform rainfall simulations for the four NBS scenarios are larger than 1 for EV1 and EV2, and around 0.8 for EV3. The reason mentioned in the previous section.

As shown in Fig. 13b, the results of $PD_{Qp}$ for PP1, RG1, and Combined1 scenarios are also consistent with the baseline scenario: $PD_{Qp}$ is the lowest for EV1, and the highest for EV2. For these three NBS scenarios, $PD_{Qp}$ range from about 8 % to 18 % for the three rainfall events. The relationship between the $SD$ of the rainfall intensity at the largest rainfall peak and the $PD_{Qp}$ of each NBS scenario (Fig. 14a) show that $PD_{Qp}$ (the uncertainty related to the peak flow) computed for PP1, RG1, and Combined1 scenarios increase simultaneously with the increase of the $SD$ of the rainfall intensity. The results computed for GR1 scenario do not depict the same tendency: $PD_{Qp}$ computed for EV3 is higher than those computed for the two other events. The reason is related to various factors. Namely, it may be affected by the intersection effects of the spatial variability of rainfall and the spatial arrangement of green roofs in the catchment. The reason can be explained by the fact that, in the GR1 scenario, the green roofs are mainly implemented on the locations with high distributed rainfall intensities. As demonstrated by many previous studies (Qin et al., 2013; Palla and Gnecco, 2015; Ercolani et al., 2018), GR are usually more effective for intense but short rainfall peaks. In the case of the GR1 scenario under the distributed rainfall of EV3, GR measures effectively stored more runoff than in the uniform rainfall during the main rainfall peak. This enlarges the variability of the hydrological response in terms of peak flow.

Regarding the percentage difference on total runoff volume, it is noticed that the computed $PD_V$ are lower than 6 % for all NBS scenarios under the three rainfall events, especially in EV3, where they are lower than 2 % (Fig. 13c). Comparing with the uncertainty on the peak flow, the resulting uncertainty on the total runoff volume is little influenced by the spatial variability of the rainfall. The reason is that the spatial variability of total rainfall depth is less pronounced with respect to the spatial variability of the rainfall intensity at the largest rainfall peak, and also there is no highly localized storm cell in studied events. Figure 3 (top) displays the rainfall intensity at the largest rainfall peak (per radar pixel) over the Guyancourt catchment for the three studied rainfall events. It is noticed that the highest rainfall peak of the distributed rainfall is very variable in space, which enlarged the discrepancy with the corresponding uniform rainfall, resulting in a significant impact on the peak flow of each NBS scenario that simulated with two different rainfall inputs. However, the cumulative rainfall of the distributed rainfall input is not very variable in space (see Fig. 3 middle). For instance, the standard deviation ($SD$) of the cumulative rainfall of the three rainfall events is around 1 mm, which indicates that the spatial variability of the distributed rainfall is not very pronounced at most of the time steps. Thus, the difference between distributed rainfall and uniform rainfall is relatively small during the whole rainfall period. Finally, the simulated flow of NBS scenarios under two different rainfall inputs is similar in most time steps, resulting in the percentage difference on the total runoff volume of NBS scenarios (simulated by distributed rainfall and uniform rainfall) is not significant.

As illustrated in Fig. 14b, the relationship between the *SD* of total rainfall depth and the $PD_V$ of NBS scenarios is nonlinear. This can be explained by the fact that the three rainfall events are relatively long, and the hydrological performances of NBS are gradually changed during the event (e.g. they can efficiently infiltrate or store water at the beginning, and be saturated after a long rainfall period). Comparing the $PD_V$ of each NBS scenario for all three rainfall events (Fig. 13c), those computed for GR1 and Combined1 appear to be the highest for EV2. It could be also related to the intersection effects of spatial location of GR measures and the spatial variability of rainfall. Indeed, these GR measures (considered in the GR1 and Combined1 scenarios) are mainly located in the north side of the catchment. In this area, the first distributed precipitation of EV2 (1-3.5 h), is relatively weak and variable (i.e., there is no rainfall or the rainfall with very low intensity in some localization pixels). Furthermore, as the initial moisture condition of GR measures are considered as unsaturated in both distributed and uniform rainfall, the GR measures are more efficient at the beginning of the distributed rainfall than in the uniform rainfall, and finally enlarge the uncertainty associated with precipitation variability (i.e., the corresponding $PD_V$). More discussion about the intersection effects is presented in Section 4.3.

## 4.3 Impacts of the spatial distribution of NBS

In order to analyze the impacts of the spatial distribution of NBS on the hydrological responses of NBS scenarios, the results of the second set of modelling experiment (described in Section 3.3) are presented as follows. As shown in Fig. 15a, the $PD_{Qp}$ of all NBS scenarios are lower than 5 %, and the $PD_V$ of all NBS scenarios are lower than 8 %, which indicates that the hydrological responses of NBS scenarios are little affected by the spatial distribution of NBS in the catchment. This result is generally consistent with the observation of Versini et al., (2016), who pointed out that the impact of the spatial distribution of green roofs on the catchment response is minimal. However, comparing the $PD_{Qp}$ of each NBS scenario, those computes for PP and GR scenarios range from about 2 % to 5 %, which are slightly higher than those related to other scenarios, especially for EV1 and EV3. The reason can be explained by two factors: (i) the infiltration or detention capacity of PP and GR measures are less effective for rainfall characterized by strong intensity and long duration (Qin et al., 2013; Palla and Gnecco, 2015), whereas the RG measures are artificial depressed green areas (simulated with a 0.3 m depression depth) with higher retention capacity (Dussaillant et al., 2004); (ii) the differences of $D_F$ (large scale; i.e., the second regime) between PP1 and PP2 scenarios as well as between GR1 and GR2 scenarios are larger than that of the other NBS scenarios (Table 4). Figure 16a shows the difference of $D_F$ between the same types of NBS scenarios is proportional to the corresponding $PD_{Qp}$. It is found that the larger the difference of $D_F$, the higher the $PD_{Qp}$ is. Regarding the $PD_V$ of NBS scenarios for the three uniform rainfall events (Fig. 15b), those comparing PP1 and PP2 scenarios (which ranges from about 4 % to 8 % for the three rainfall events, especially higher for the two strong and long events) are slightly higher than those related to the other scenarios. Because porous pavements are infiltration-based measures that gradually discharging water into the underlying layers, their performances are more related to the heterogeneity of their performed location. Namely, some PP measures implemented in drained areas may suffer more from surface runoff, are therefore more easily saturated (see Fig. 6 for a comparison of the spatial arrangement of PP measures for two PP scenarios). As shown in Fig. 16b, the difference of $D_F$ between the same types

of NBS scenarios has a moderate positive correlation ($r =0.61$) with the corresponding $PD_V$. Our study hypothesizes that this rather weak correlation is related to the complexity of rainfall with several peaks and dry periods, the retention/infiltration capacity of NBS changes with the rainfall intermittency.

**4.4 Intersection effects of spatial variability of rainfall and spatial arrangement of NBS**

In the following, we present the results of the third modelling experiment set described in Sect. 3.3. The aim is to analyse the potential intersection effects of spatial variability of rainfall and spatial distribution of NBS on the hydrological responses of NBS scenarios.

The resulting uncertainty on the peak flow and total runoff volume ($PD_{Qp}$ and $PD_V$) of the third set of modelling experiments are shown in Fig. 17. Firstly, we found that the spatial variability of rainfall has a certain extent impact on the peak flow of

550 each scenario, with the $PD_{Qp}$ ranging from about 8 % to 18 %. With the exception of GR1, all the NBS scenarios have a similar tendency: the $PD_{Qp}$ are the lowest for the first event, and the highest for the second one. Namely, for most of NBS scenarios, the $PD_{Qp}$ (uncertainty on peak flow) increases with the increase of the spatial variability of rainfall intensity. As shown in Fig. 17c, comparing the $PD_{Qp}$ between scenarios of PP1 and PP2, RG1 and RG2, as well as Combined1 and Combined2 for the three rainfall events, the maximum difference is less than 3 %. However, comparing the $PD_{Qp}$ between GR1 and GR2, the

555 difference is larger, especially in EV3 (> 6 %). For the GR1 scenario, $PD_{Qp}$ range from about 8.7 % to 18 % in all three rainfall events, and those of GR2 range from about 10.7 % to 16 %. Furthermore, for GR1, the largest $PD_{Qp}$ is in EV3, but for GR2, the largest $PD_{Qp}$ is computed for EV2. The difference of $PD_{Qp}$ between GR1 and GR2 scenarios demonstrated that the spatial variability of rainfall and the spatial arrangement of GR measures have some intersection effects on the peak flow of GR scenarios. However, it is not evident for the other NBS scenarios. One of the reasons has been discussed in Sect. 4.2.2: in the

560 GR1 scenario, GR measures are mainly implemented in the north part of the catchment, which coincidently received higher rainfall (distributed EV3); namely, the "hot spots" of the rainfall field were highly intersected by the GR measures due to their high fractal dimension. Therefore, the peak flow was effectively reduced by the GRs. On the contrary, for GR2 scenario, the GR measures are mainly located on the south side of the catchment, which scarcely intersected with the rainfall spikes. Thus, comparing with the GR1 scenario, the difference of GR2 scenario simulated under the distributed rainfall and uniform rainfall

is less significant. Another possible reason is GR has the lowest storage capacity in the studied NBS, as well as the studied rainfall events are not intense enough to saturate the other types of NBS (see Versini et al., 2016 for a comparison of different properties of GR). Her et al. (2017) also indicated the hydrological performances of NBS are sensitive to their configurations. However, the most plausible reason is that the intersection effect is more perceptible for GRs, as they only respond to local precipitation, while it is often masked for other NBS measures that must also mitigate runoff received from other parts of the

watershed. Indeed, the already mentioned integrative character of runoff should reduce the evidence for intersection effects in other NBS scenarios, whether for distributed or uniform rainfall. Similarly to Fig. 13, Fig. 17 demonstrates the percentage difference on peak flow which is much higher than that of the total runoff for each scenario. The reason is the same as explained in Sect. 4.2.2.

Concerning the intersection impact on total runoff volume of NBS scenarios, the variations of $PD_V$ among most of NBS scenarios pairs (PP1 and PP2, GR1 and GR2, as well as Combined1 and Combined2) are significantly different for the three rainfall events. The maximum discrepancy (around 5 %) is found between Combined1 and Combined2 in EV3. Indeed, the NBS can effectively reduce the water volume until their saturation, in particular when they largely intersect with higher rainfall. Lower intersect results in higher simulated flows and longer transfers. Furthermore, the cumulative distributed rainfall is more variable for EV3. Conversely, the difference of $PD_V$ between RG1 and RG2 is relatively small, which is less than 1 %. The reason can be explained by the large retention capacity of RG measures, which has been mentioned in Sect.4.3.

To further investigate the intersection effects, the fourth subset of modelling experiment is used. As shown in the hydrographs (Fig. 18), the peak flow of GR1 scenario was expected to be less than that of GR2, and this is confirmed for EV4, EV5, and EV6. For EV4 and EV5, with the same maximum intensity of 55 mm/h, the hydrographs of these two events significantly differ, with the peak flow decreasing by a factor 2 for EV5. However, the only difference in the rainfall inputs is that there is zero rainfall outside of the GRs during the 3 min rainfall peak. The $PD_{Qp}$ and $PD_V$ of GR1 and GR2 scenario under the EV4 is around 5 %, and 4.3 %, respectively (see Fig. 19). For EV5, the $PD_{Qp}$ and $PD_V$ increase to 20.7 % and 7.8 %, respectively. This confirms that without the impact of runoff that generated by other land uses, the intersection effects increase considerably with the higher spatial variability of rainfall intensity. For the EV6, the maximum rainfall intensity during the 3 min has been decreased to 17 mm/h. This was sufficient to further reduce the peak flow during the principal rainfall peak. For this event, the $PD_{Qp}$ and $PD_V$ values drop to 3.5 % and 1.8 %, respectively. This indicates that the intersection effects is less significant for the rainfall with the lower spatial variability of rainfall intensity. As expected in the EV7 scenario, because of zero rainfall intersected with the GRs in GR1 scenario, the peak flow of GR2 remains slightly lower than that of the GR1, with the $PD_{Qp}$ and $PD_V$ values of only 2.1 % and 1.4 %, respectively.

Overall, the results demonstrate that the spatial variability of rainfall and the spatial arrangement of NBS can generate uncertainties on peak flow and total runoff volume estimations if they are not considered properly. This suggests that the performances of NBS scenarios that evaluated by some studies with only applying uniform rainfall as input can be biased in terms of the intersection effects (Zahmatkesh et al., 2014; Ahiablame et al.,2016; Guo et al., 2019). In our specific case, the intersection effect is more significant for GR scenarios and combined scenarios in terms of peak flow and total runoff volume, respectively. However, the physical properties of NBS are indeed another significant factor for the overall performances of scenario (Gilroy et al., 2009), for example, the intersection effect is less evident for RG scenarios mainly due to their high storage capacity. Comparing to the impacts of spatial variability of rainfall on the hydrological responses of NBS, the intersection effects seem to be less significant. This results also further demonstrated the hydrological responses of NBS scenario is less influenced by the spatial distributions of NBS. As the rainfall fields are always variable in space and time, to make the most of the benefits of NBS for stormwater management, the results suggest implementing NBS scattered in the catchment, but with a higher fractal dimension $D_F$. This will combine a lower investment with the maximum return, preventing NBS from concentrated in certain specific places.

**5 Conclusions**

This paper studies the uncertainty of the hydrological responses of Nature-Based Solutions (NBS) scenarios resulting from the multi-scale spatial variability of rainfall and heterogeneous distribution of NBS at the urban catchment scale. As an application of the multifractal approach, we pointed out how the "multifractal intersection theorem" can quantify how often they intersect, which conditions the performance of NBS. The high-resolution distributed rainfall data from the École des Ponts ParisTech (ENPC) X-band radar depict the spatially variable rainfall fields. The fully-distributed and physically-based hydrological model (Multi-Hydro) takes into account the heterogeneity of an urban environment down to the 10 m scale, including the spatial arrangement of NBS and spatial distribution of rainfall. The principal findings are summarized as follows:

1. The spatial variability of rainfall has a significant impact on the peak flow of NBS scenarios for the three studied rainfall events. For instance, it makes the maximum percentage difference on peak flow ($PD_{Qp}$) increase up to 18 % in GR1 scenario. Furthermore, the spatial variability of the rainfall intensity at the largest rainfall peak is almost linearly related to the $PD_{Qp}$ computed for all NBS scenarios (except for GR1): the more variable are the rainfall intensities, the higher are the $PD_{Qp}$. However, the resulting percentage difference on total runoff volume ($PD_V$) computed for all NBS scenarios show that the spatial variability of rainfall has much lower impact on the uncertainty related to total runoff volume: the average $PD_V$ being of the order of 2.3 % only.

2. The impact of spatial arrangement of NBS on hydrological responses of the catchment is less obvious. For all the NBS scenarios, $PD_{Qp}$ and $PD_V$ are lower than 5 % and 8 %, respectively. However, we found that the difference of fractal dimension ($D_F$) between the same types of NBS scenarios has a fairly strong positive correlation to the related $PD_{Qp}$. Therefore, we suggest implementing NBS by optimizing $D_F$ over the whole catchment to be the highest possible. Furthermore, mixing different NBS in the catchment, as presented in the two combined scenarios, can also efficiently reduce the uncertainty associated with the spatial arrangement of NBS.

3. The fractal dimension $D_F$ appears as a useful tool to quantify the spatial heterogeneity of NBS across a range of scales. The $D_F$ of each NBS scenario is associated with the urbanization level of the catchment, which confirms that the level of implementation of NBS is reasonable to match the catchment conditions. The fractal dimension combined with the fully-distributed model is an innovative approach that is easily transportable to other catchments.

4. The spatial distribution of rainfall and the spatial arrangement of NBS have intersection effects on the hydrological responses of NBS scenarios, especially significant for the peak flow of green roof (GR) scenarios (with a maximum difference between the scenario of GR1 and GR2 reaching about 6 % on peak flow). The intersection effects on the total runoff volume of each NBS scenario is quite variable because the chosen NBS present some limitations in terms of infiltration or detention capacity during a long rainfall event with high intermittency. However, the rain garden (RG) scenarios appear to be less affected by the intersection effects, with a difference lower than 3 % on peak flow and lower than 1 % on total runoff volume, mainly due to RG measures characterised with higher retention capacity. The results of the synthetic experiment firstly confirm that there is a complex interplay between the spatio-temporal intensity of precipitation and the runoff received from other parts of the watershed. Furthermore, this experiment strengthened the

intersection effects on the GR scenarios. These intersection effects can be more significant for the rainfall events with higher spatial variability.

5. The study of hydrological response in various NBS scenarios resulting from the multi-scale spatial variability of precipitation and the heterogeneous distribution of NBS hints towards using fully distributed hydrological models over semi-distributed or lumped models. Indeed, the fully distributed model has been shown to be able to take into account these small-scale heterogeneities and propagate their effects to watershed scales, while parameterizing or smoothing out some critical heterogeneity, as done in non-fully distributed models, may bias its predictions.

In our specific case, the GR scenarios are more sensitive to the spatial variability of rainfall and the spatial arrangement of GR measures, while the performances of RG scenarios and combined scenarios are more stable under any condition. Apparently, these findings already give some incites to decision-makers on *Why* they need to prioritize given NBS within the urban planning process.

Although the rainfall events selected for this study were not extreme events, they cover a rather broad spectrum of spatio-temporal variability in rainfall, and they are very typical precipitations in the Paris region. The simulation results can serve as a reference for future urban planning in this region. For example, the results of three different impacts (i.e., the spatial variability of precipitation, the spatial distribution of NBS, and the intersection effects) on the performance of NBS scenarios are useful for decision-makers, targeting for an actual project.

However, larger precipitation samples, including extreme rains, as well as NBS monitoring data will be helpful to get a better knowledge of somehow universal solutions and provide answers on *How* to prioritize these NBS. With respect to this perspective, the obtained results already demonstrated that new scale-independent indictors, like the fractal dimension applied in this study, will be essential for more profound quantitative evaluation of the diversity of combined impacts, including for other heterogeneous catchments. Therefore, this study has an important potential impact, due to its originality with respect to the nonlinear tools used to address such practical issues, and its relevance in interdisciplinary applications. This suggests to pursue the development of original tools to get new insights into the scaling complexity of flows in urban hydrology.

*Data availability*. Datasets for this study are under preparation for public release by the Chair Hydrology for Resilience Cities. They can already be freely obtained via contact.hmco@enpc.fr.

*Model availability*. The version 2.1 of Multi-Hydro is registered by the (France) Program Protection Agency (IDDN FR 001 340017 000 SC 2015 0000 31235) and can be freely obtained via contact.hmco@enpc.fr.

*Author contribution*. Yangzi Qiu performed the modelling experiments, data analysis and wrote the paper with contributions from all co-authors. Ioulia Tchiguirinskaia, Daniel Schertzer, conceived and supervised the study. Yangzi Qiu and Pierre-

Antoine Versini created and implemented various scenarios. Yangzi Qiu and Igor Paz prepared the input data for the model. All co-authors revised the manuscript.

*Competing interests.* The authors declare that they have no conflict of interest.

*Acknowledgements.* The first author greatly acknowledges the financial support by the China Scholarship Council. The first author furthermore highly acknowledges Abdellah ICHIBA for his precious help in taking over the Multi-Hydro model. All the authors acknowledge the Chair "Hydrology for Resilient Cities" (endowed by VEOLIA), for partial financial support and
680 express gratitude to their colleagues from SIAVB and Veolia, particularly Bernard WILLINGER, for many stimulating discussions. The authors acknowledge the agglomeration community of Saint-Quentin-en-Yvelines for providing the data of the Guyancourt catchment; and express thanks to Aderson Dionísio Leite Neto for his help with numerical pre-processing of these data, as a part of his secondment at École des Ponts ParisTech.

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

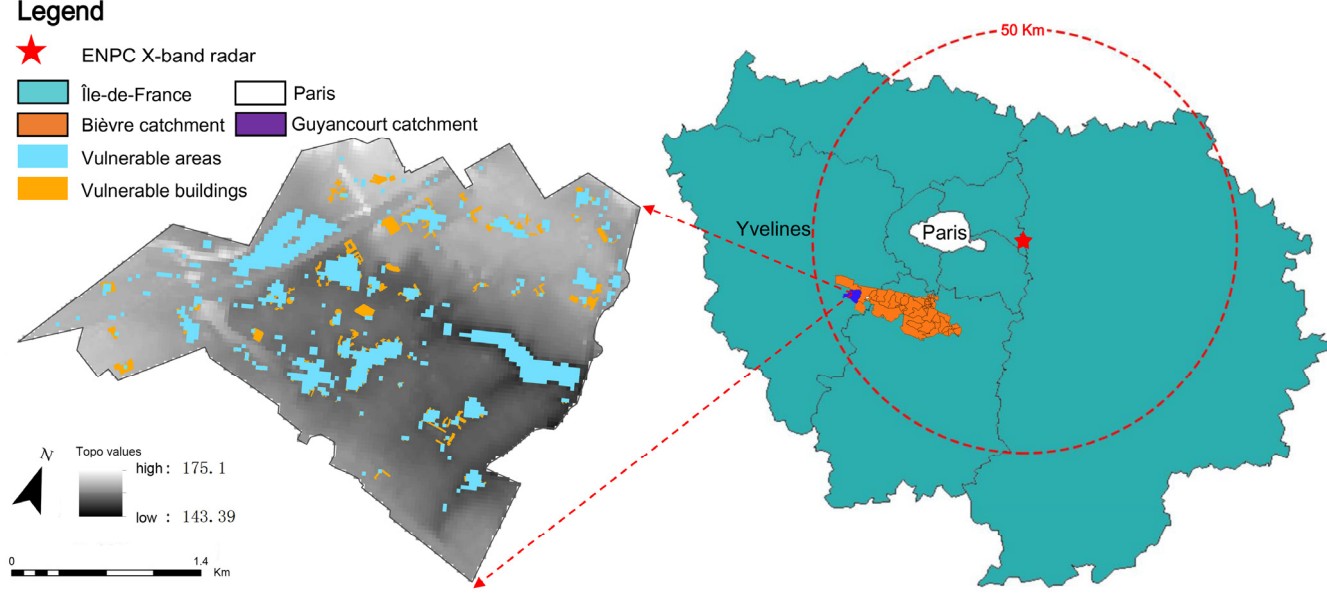

**Figure 1: Location of the study site and the corresponding topography map, highlighting some vulnerable areas and buildings at risk of waterlogging in the Guyancourt catchment.**

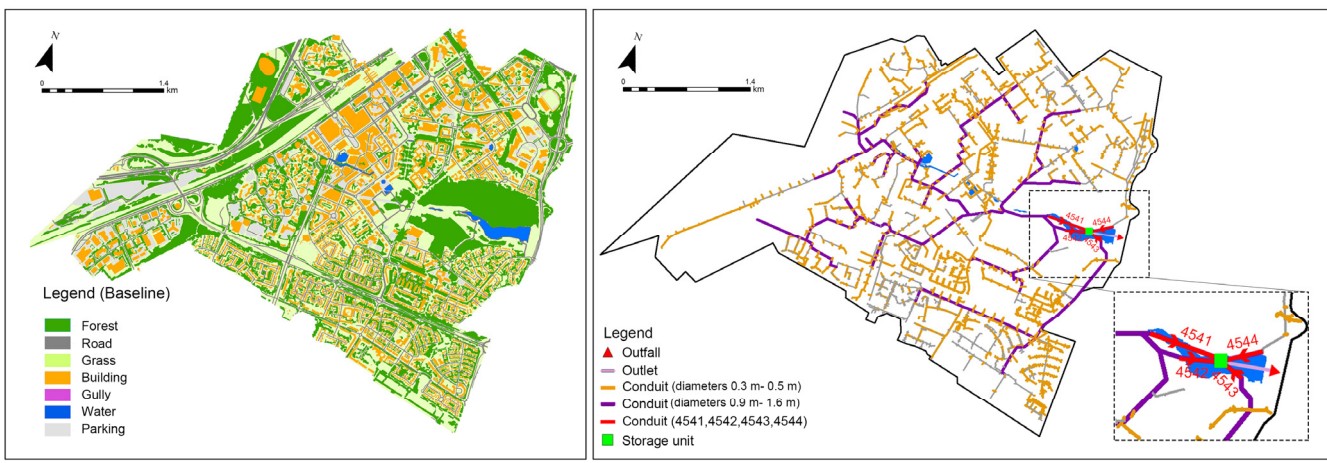

**Figure 2: Left: land use map (baseline scenario);right: drainage system with four conduits (4541, 4542, 4543, and 4544) highlighted.**

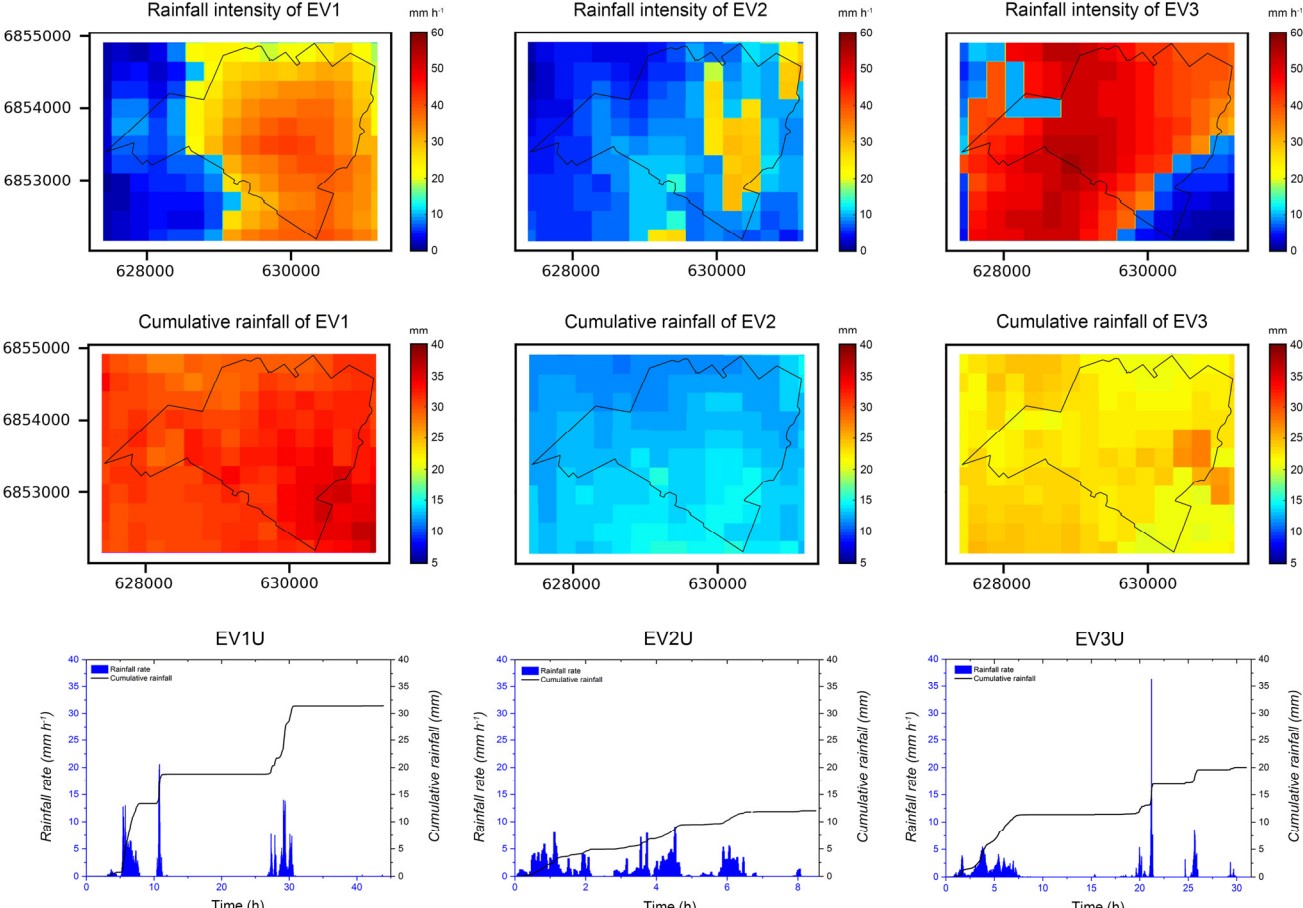

**Figure 3: Top: The rainfall intensity at the largest rainfall peak (per radar pixel) over the Guyancourt catchment area for the three studied rainfall events; middle: cumulative rainfall depths (per radar pixel) over the Guyancourt catchment area for the three studied rainfall events; bottom: time evolution of rainfall rate (mm h⁻¹) and cumulative rainfall (mm) of the three uniform rainfall events over the whole catchment.**

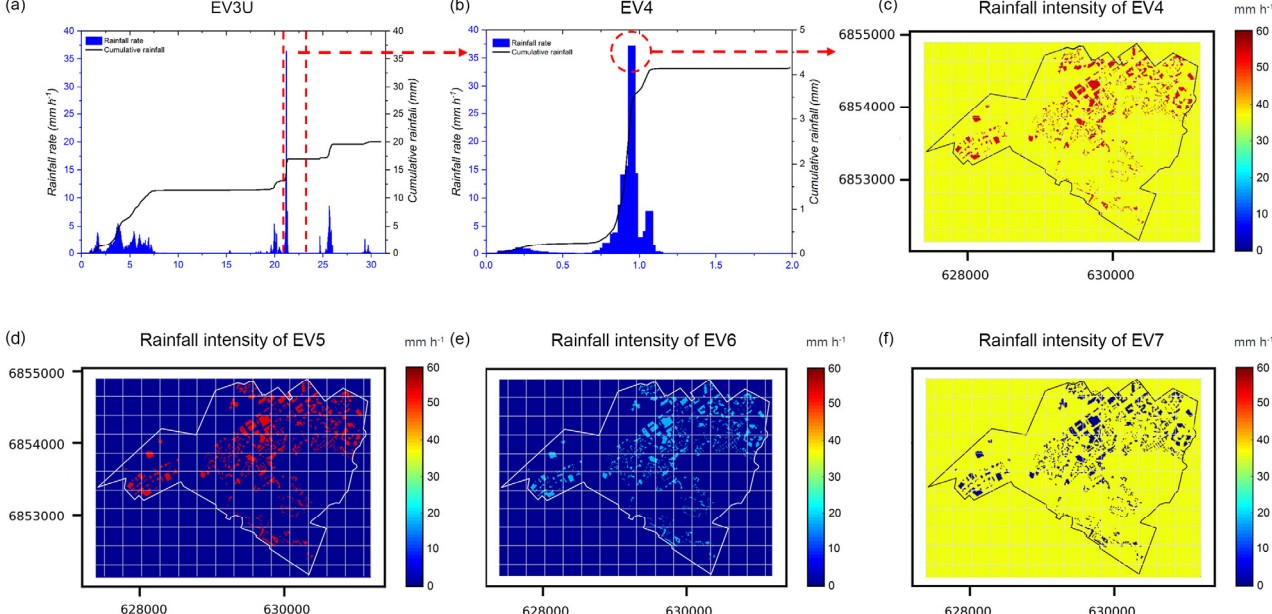

Figure 4: (a) Time evolution of rainfall rate (mm h$^{-1}$) and cumulative rainfall (mm) of the EV3U over the whole catchment (the period between the red dash lines is the selected period for creating the EV4); (b) time evolution of catchment-averaged rainfall rate and cumulative rainfall of the EV4 over the whole catchment; (c) the rainfall intensity at the largest rainfall peak (distributed) over the Guyancourt catchment for the EV4 (the red pixels are the location of GRs in GR1 scenario with the highest rainfall intensity in space), the rainfall of other areas are uniform; (d-f) the rainfall intensity at the largest rainfall peak (distributed) over the Guyancourt catchment for the EV5, EV6 and EV7, respectively.

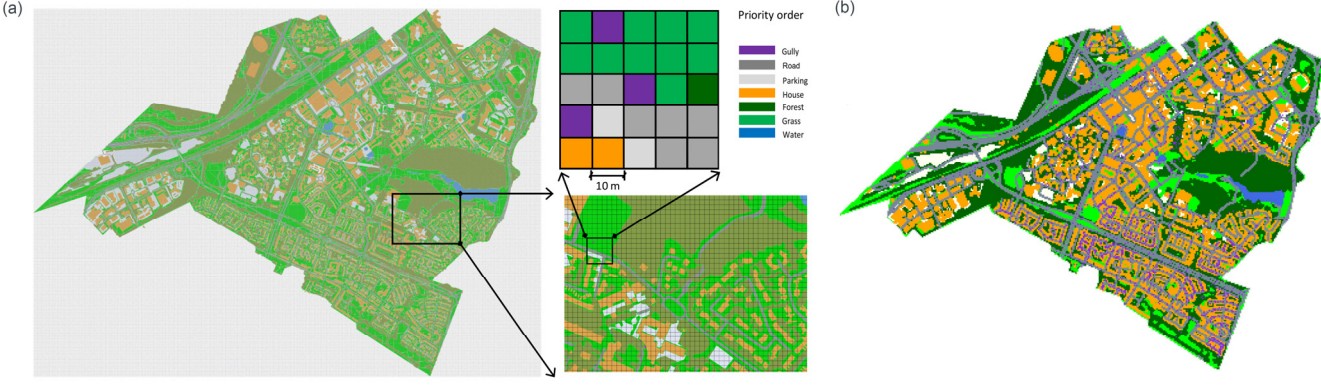

**Figure 5: (a) Rasterization of the original land use data into 10 m with priority order, and (b) the rasterized land use data.**

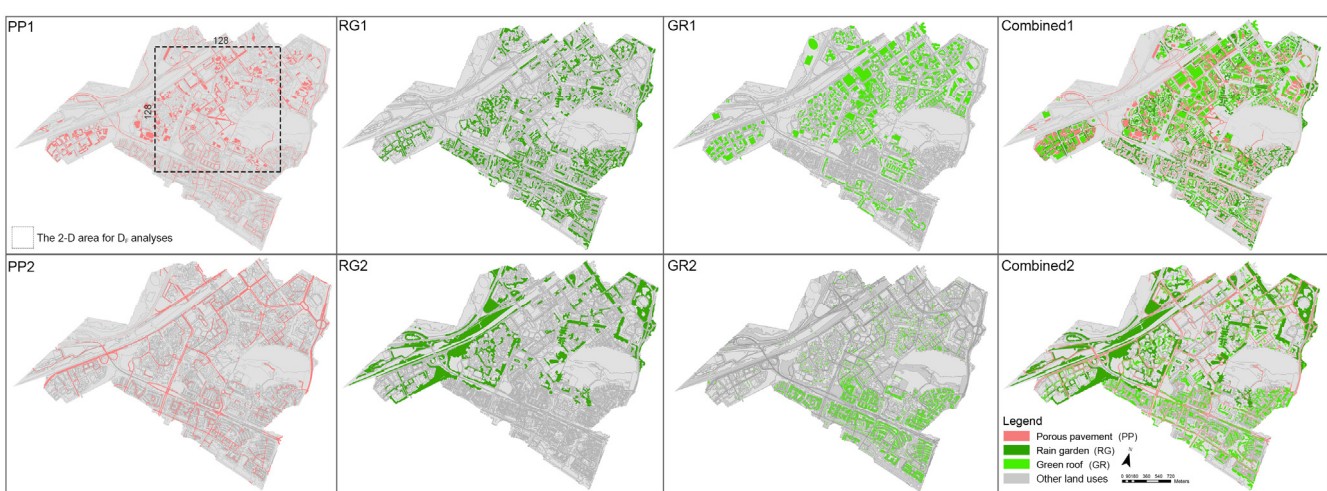

**Figure 6: Two scenarios for each of NBS implementation, including porous pavement (PP1, PP2), rain garden (RG1, RG2), green roof (GR1, GR2), Combined1 and Combined2, the rectangular area that presented in the PP1 scenario is the example area for applying fractal analysis.**

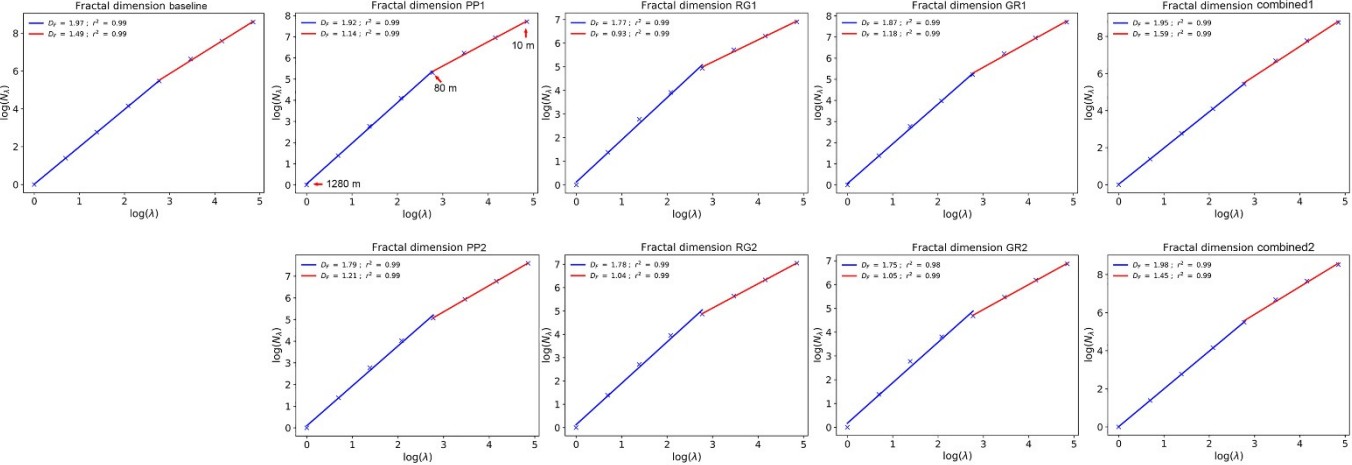

**Figure 7: The fractal dimension of impervious surface of the baseline scenario and the fractal dimension of NBS in each NBS scenario.**

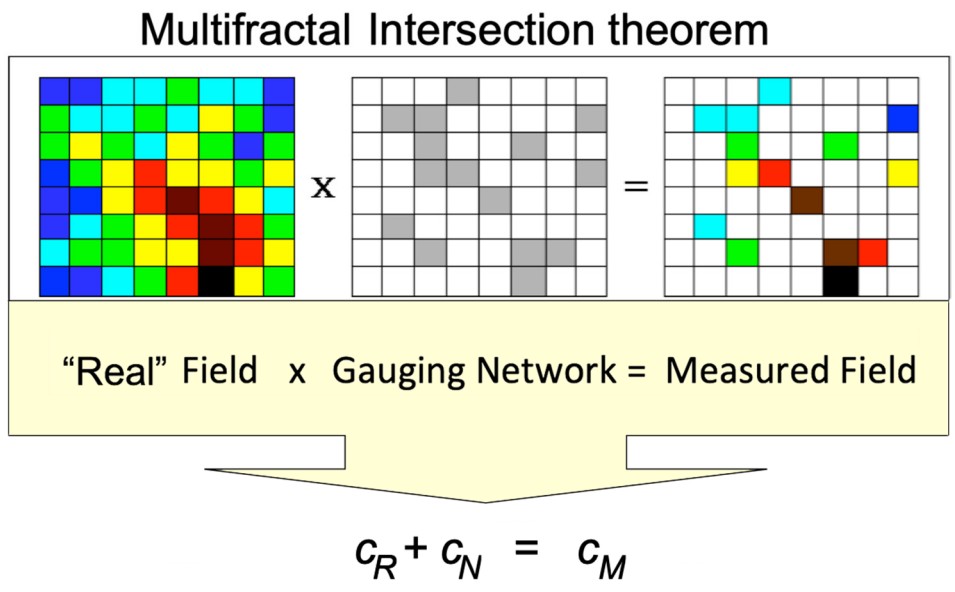

**Figure 8: Schematic of the (multifractal) intersection theorem applied to the measured rainfall M by a rain gauge network N. The measured rainfall corresponds to the product of the "real" rainfall R by the gauge characteristic function (=1 if there is a gauge in this pixel, 0 otherwise) and the corresponding codimensions $c_R = d - D_R$ and $c_N = d - D_N$ add to yield the codimension of the measured rainfall $c_M = d - D_M$; d is the embedding space dimension, DR, DN and DM are the corresponding fractal dimensions (adapted from Tchiguirinskaia et al., 2004).**

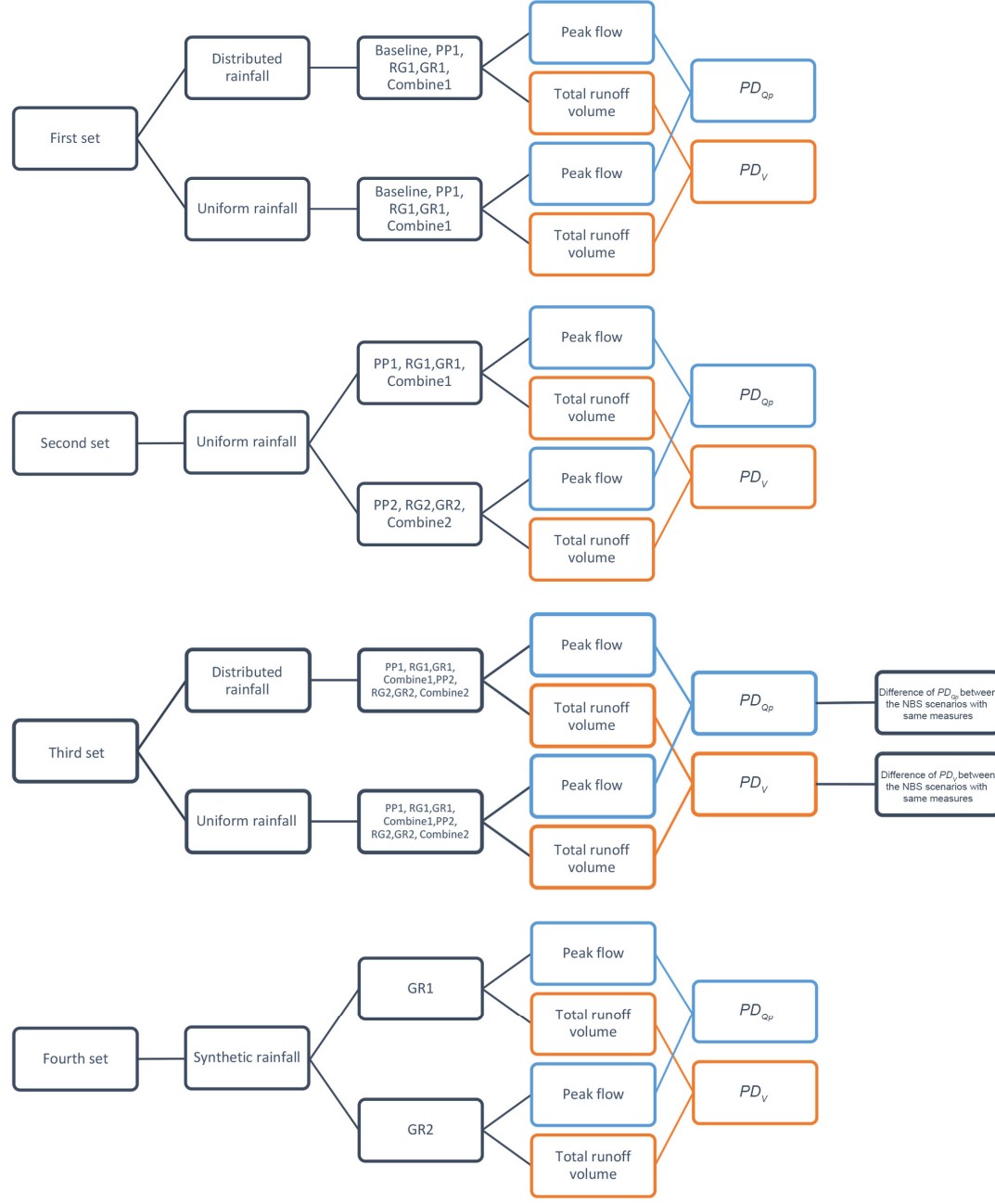

**Figure 9: Flow chart of the four sets of modelling experiments.**

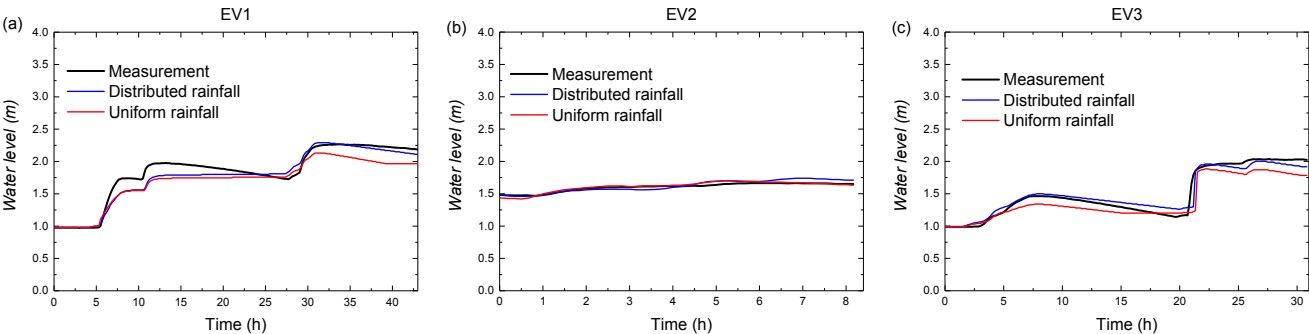

**Figure 10: Comparison of the observed and simulated water levels (simulated with distributed rainfall and uniform rainfall) of the three rainfall events: (a) EV1, (b) EV2, (c) EV3.**

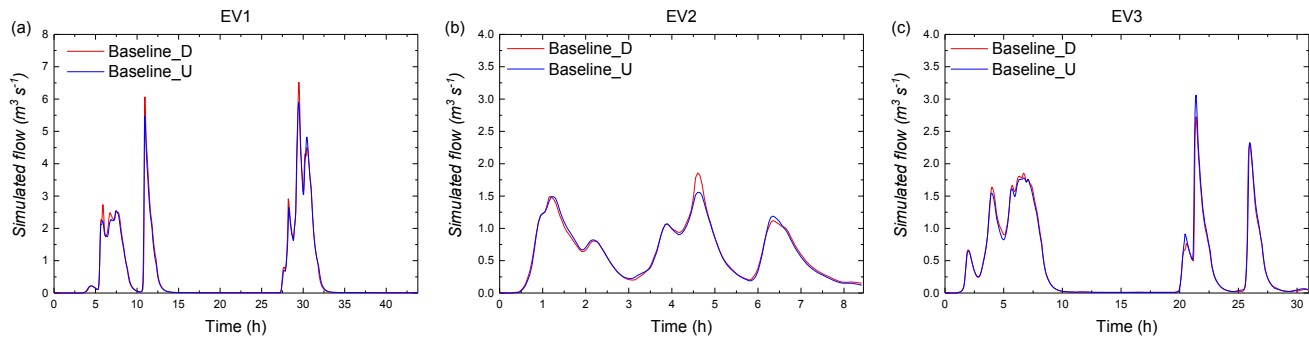

**Figure 11: Simulated flow (m³ s⁻¹) of the baseline scenario under three distributed rainfall events and three uniform rainfall events: (a) EV1, (b) EV2, (c) EV3.**

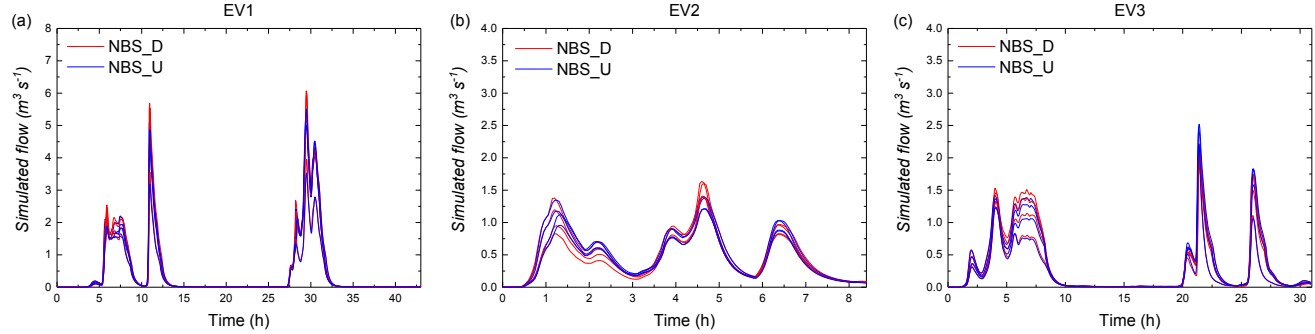

**Figure 12. Simulated flow (m³ s⁻¹) of the first set of NBS scenarios under three distributed rainfall events and three uniform rainfall events (the red hydrographs represent the NBS scenarios simulated with distributed rainfall, and the blue hydrographs represent the NBS scenarios simulated with uniform rainfall): (a) EV1, (b) EV2, (c) EV3.**

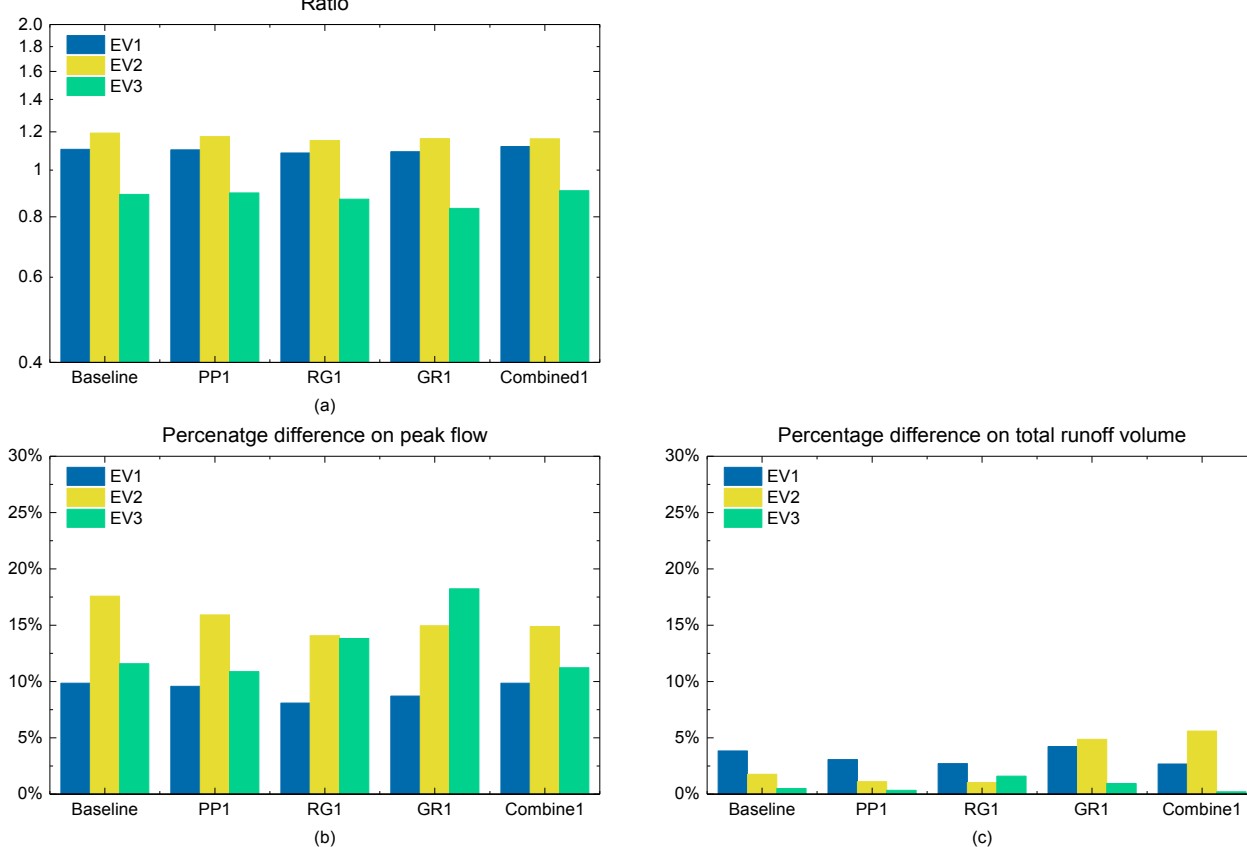

**Figure 13. (a) The ratio of peak flow between the scenarios under the distributed rainfall and the scenarios under the uniform rainfall; (b) percentage difference on peak flow of the baseline scenario and the first set of NBS scenarios under the three distributed rainfall events and the three uniform rainfall events; (c) percentage difference on total runoff volume of the baseline scenario and the first set of NBS scenarios under the three distributed rainfall events and the three uniform rainfall events.**

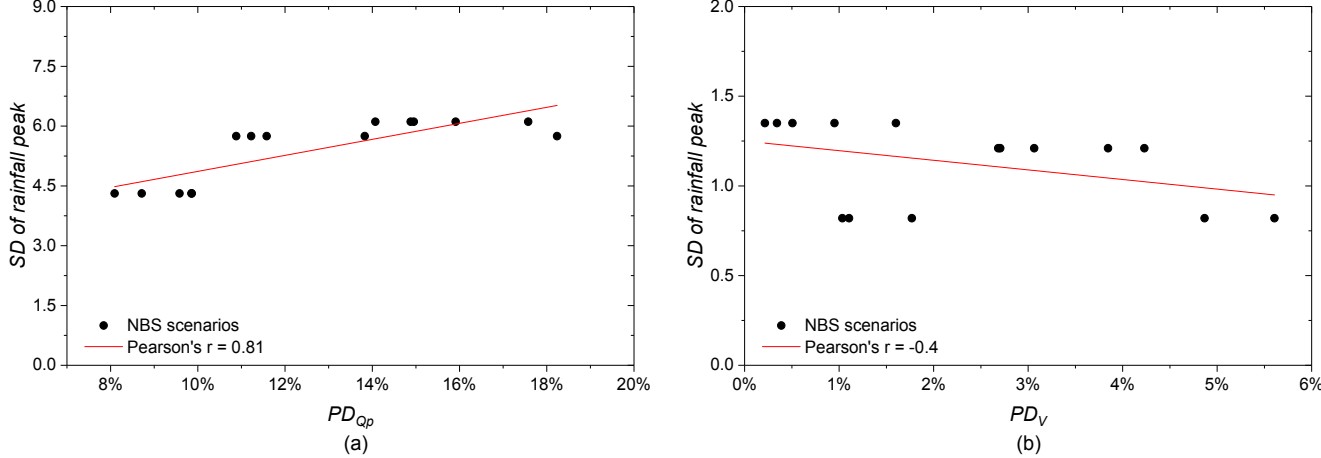

**Figure 14. (a) Relationship between the *SD* of rainfall intensity at the largest rainfall peak and *PD$_{Qp}$* of NBS scenarios; (b) relationship between the *SD* of total rainfall depth and *PD$_V$* of NBS scenarios.**

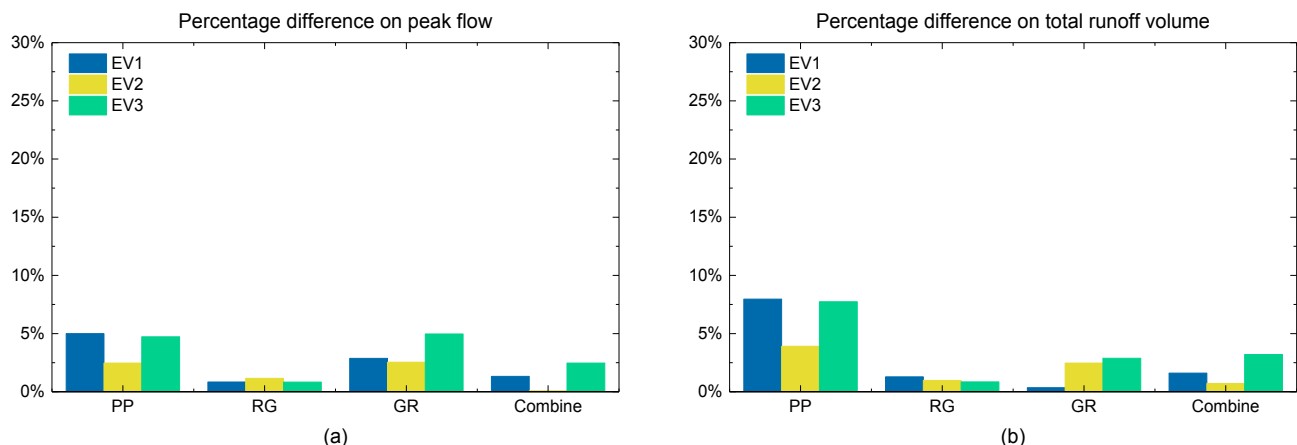

**Figure 15. (a) Percentage difference on peak flow between the same types of NBS scenarios under the three uniform rainfall events; (b) percentage difference on total runoff volume between the same types of NBS scenarios under the three uniform rainfall events.**

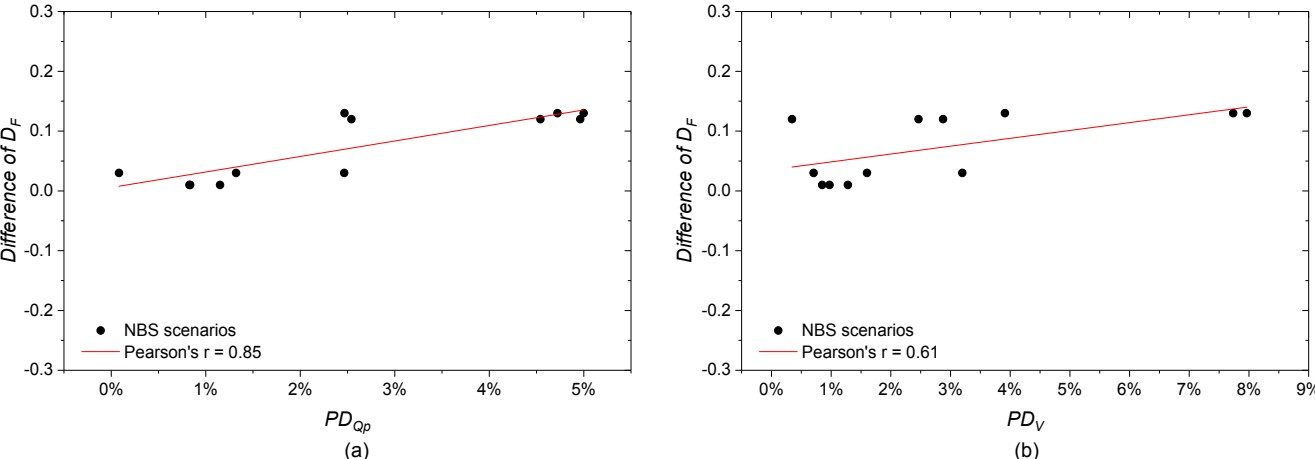

**Figure 16. (a) Relationship between the difference of $D_F$ of the same types of NBS scenarios and $PD_{Qp}$ of the same types of NBS scenarios; (b) relationship between the difference of $D_F$ of the same types of NBS scenarios and $PD_V$ of the same types of NBS scenarios.**

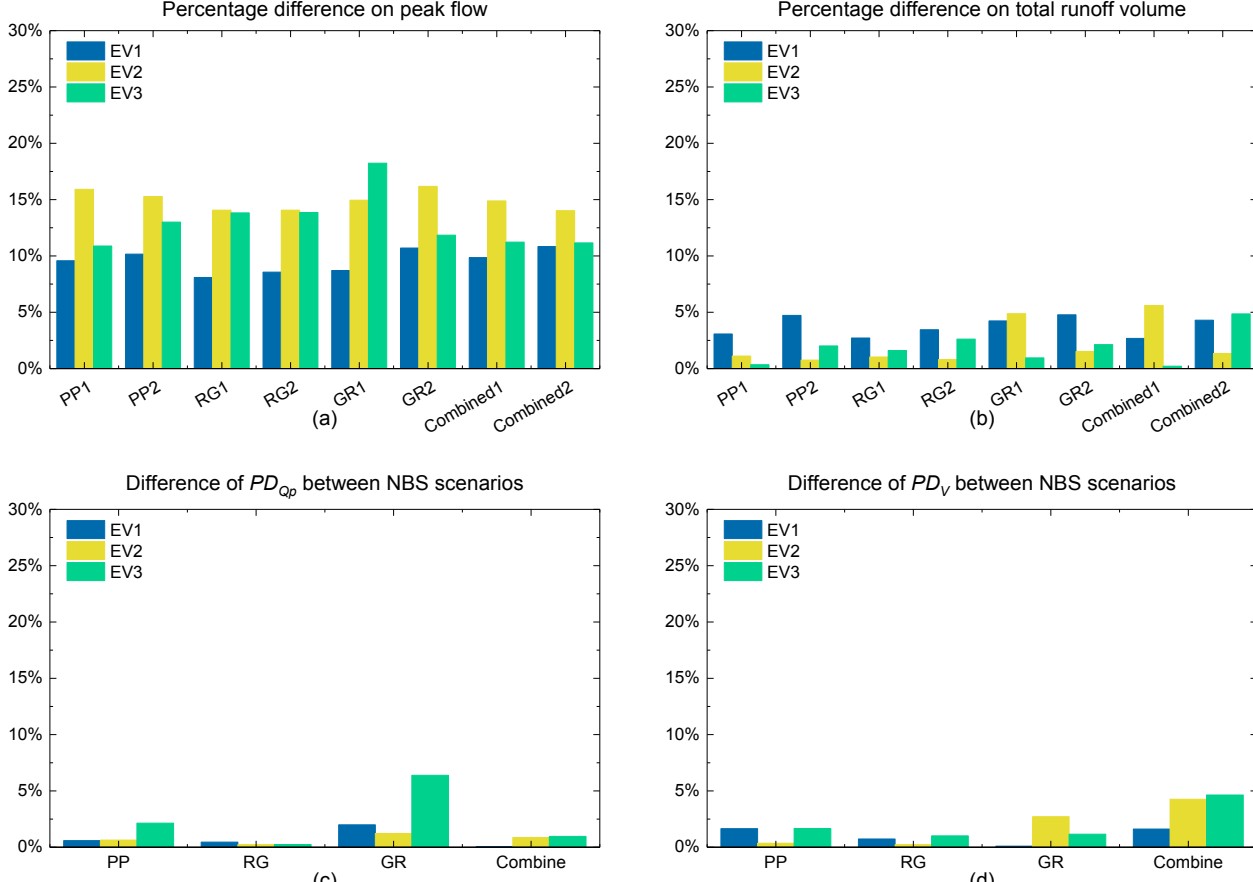

**Figure 17. (a) Percentage difference on peak flow of all NBS scenarios under the three distributed rainfall events and the three uniform rainfall events; (b) percentage difference on total runoff volume of all NBS scenarios under the three distributed rainfall events and the three uniform rainfall events; (c) difference of $PD_{Qp}$ between the same types of NBS scenario. (d) Difference of $PD_V$ between the same types of NBS scenario.**

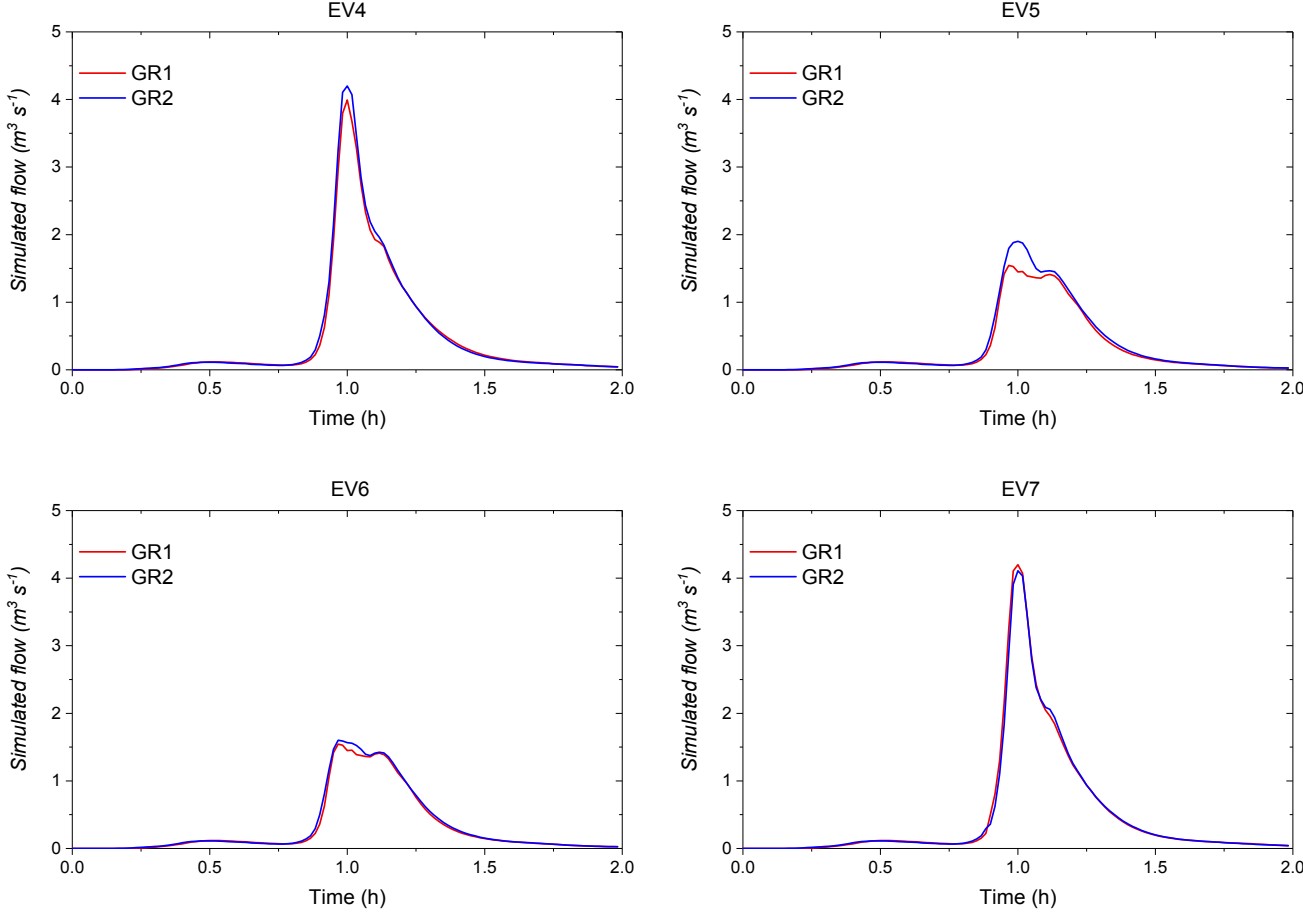

**Figure 18.  Simulated flow (m³ s⁻¹) of GR1 and GR2 scenarios under the four syntactic rainfall events.**

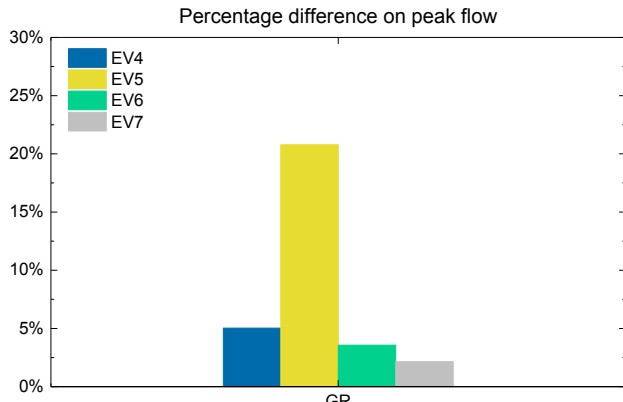
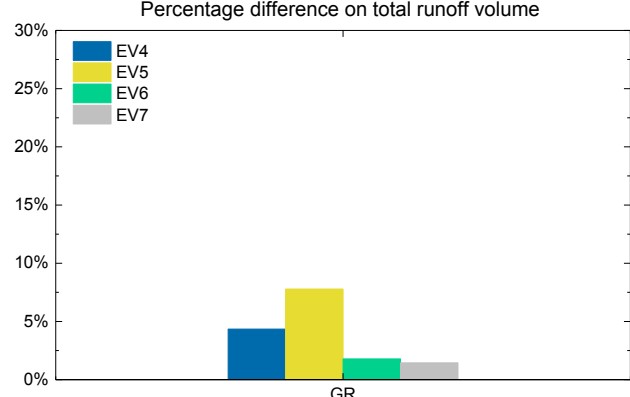

**Figure 19. (a) Percentage difference on peak flow of GR scenarios under the four syntactic rainfall events; (b) percentage difference on total runoff volume of GR scenarios under the four syntactic rainfall events.**

**Table 1. Main characteristics of selected rainfall events and standard deviation (*SD*) of the rainfall intensity at the largest rainfall peak and the total rainfall depth of the three rainfall events.**

| Event ID | EV1 | EV2 | EV3 |
|---|---|---|---|
| Data | 12-13/09/2015 | 16/09/2015 | 05-06/10/2015 |
| Duration (h) | 44 | 8.4 | 31 |
| Total depth (mm) (areal average/pixel min/pixel max) | 31.5/27.4/36.9 | 12/10.43/14.1 | 20/17.6/25.4 |
| Max intensity (mm h$^{-1}$) over 1 min (areal average/individual pixel) | 20.5/41.2 | 9/29.1 | 36.4/55.6 |
| *SD* of rainfall intensity at the largest rainfall peak (mm h$^{-1}$) | 4.31 | 6.11 | 5.75 |
| *SD* of total rainfall depth (mm) | 1.21 | 0.82 | 1.35 |

**Table 2. Hydrological parameters for each land use class.**

| Land use | Manning's coefficient (no units) | Hydraulic conductivity ($m\ s^{-1}$) | Interception (mm) |
|---|---|---|---|
| Impervious surfaces (road, house, parking …) | 0.012 | 1.0e-10 | 1.9 |
| Gullies | 0.9 | 1.0e-0 | 0 |
| Grass | 0.15 | 1.9e-6 | 3.81 |
| Forest | 0.8 | 1.9e-6 | 7.62 |
| Water | 0.9 | 1.0e-0 | 100 |
| Porous pavement | 0.014 | 1.0e-4 | 2.14 |
| Rain garden | 0.2 | 1.9e-5 | 7.62 |
| Green roof | 0.14 | 3.3e-4 | 3.81 |

**Table 3. The statistical comparison of DEM5-10 and DEM25-10.**

| Statistic metrics | DEM25-10 | DEM5-10 |
|---|---|---|
| Median | 143.3 | 143.4 |
| Mean | 160.1 | 160.1 |
| Maximum | 175.4 | 175.9 |
| Minimum | 143.0 | 143.3 |
| Standard deviation | 80.2 | 80.2 |
| Root Mean Square Error | 0.26 | |
| Correlation coefficient | 0.99 | |
| Maximum difference | 5.3 | |
| Mean difference | 0.01 | |

**Table 4. The details of simulation: NBS scenarios.**

| NBS measure | Scenario | Proportion of implementation in whole catchment / selected area (after rasterization) | $D_F$ of NBS in small scale/ large scale (after rasterization) | Description of scenario |
|---|---|---|---|---|
| Porous pavement (PP) | PP1 | 8.0 %/13.8 % | 1.14/1.92 | Porous pavements were implemented on the non-driveways (width equal and less than 2.5 m) and some parking lots. |
| | PP2 | 8.0 %/10.1 % | 1.21/1.79 | Porous pavements were implemented on secondary driveways (width between 2.5 m to 5 m). |
| Rain garden (RG) | RG1 | 8.2 %/6 % | 0.93/1.77 | The low elevation greenbelts around houses were implemented by rain gardens, which can collect and store up the surface runoff from surrounding impermeable areas before infiltration on site. When rain garden saturated, the redundant surface runoff will drain into the drainage system. |
| | RG2 | 8.2 %/7 % | 1.04/1.78 | The low elevation greenbelts around public buildings and parking lots. |
| Green roof (GR) | GR1 | 8.6 %/13.5 % | 1.18/1.87 | Small and light green roofs consisting of a soil layer and a storage layer are implemented on all flat roofs. |
| | GR2 | 8.6 %/6 % | 1.05/1.75 | Impervious roofs with slightly slope ($\leq 15°$) were converted to small and light green roofs (Stanic et al., 2018). |
| NBS combinations | Combined1 | 24.8 %/38.5 % | 1.59/1.95 | A combination of PP1, RG1, GR1 |
| | Combined2 | 24.8 %/30.4 % | 1.45/1.98 | A combination of PP2, RG2, GR2 |

**Table 5.** *NSE* coefficients and *PE* values of baseline scenario under the three distributed rainfall events and three uniform rainfall events.

| Event ID | Distributed rainfall | | Uniform rainfall | |
| --- | --- | --- | --- | --- |
| | *NSE* | *PE* (%) | *NSE* | *PE* (%) |
| EV1 | 0.926 | 4.6 | 0.824 | 7.9 |
| EV2 | 0.929 | 2.2 | 0.948 | 1.96 |
| EV3 | 0.954 | 3.9 | 0.865 | 6.9 |

