# Peer review of "Space variability impacts on hydrological responses of Nature-Based Solutions and the resulting uncertainty: a case study of Guyancourt (France)"

_Hydrology and Earth System Sciences, 2020_

## Referee Comment (RC1) · Anonymous Referee #1 · 8 Jan 2021

In this article, the authors aim to investigate the uncertainty of hydrological responses in various NBS scenarios resulting from the spatial variability of rainfall and the heterogeneous distribution of NBS at the urban catchment scale. I find the manuscript to be quite suitable for HESS and presents a straight forward method which is intended to provide means to modelling works for NBS at the urban scale.

Overall, the manuscript is well presented. I would like to suggest a Minor revision for this paper, but there are a few points that should be addressed to improve its quality. 1. You use 25-m resolution DEM but the model was implemented with a 10 m spatial resolution, could you please make a comment or discussion about this? Moreover, 25-m resolution DEM seems relatively quite rough to use for an urban area. Do you think this is the limitation of your study? If you use a more detailed DEM, will it impact

your result? 2. Figure 5, In the legend it should include an abbreviation of each NBS measures e.g., Porous pavement (PP), Rain garden (RG),... 3. Please consider the results of validation from Line 328 to the Results section as this is results of the validation of your baseline scenario. 4. For Figure 12 and 16, can you discuss more on why percentage errors of peak flow are much higher than total runoff volume
* * *

---

## Referee Comment (RC2) · Ruben Imhoff (Referee) · 27 Jan 2021

**1   Summary**

The manuscript describes the effect of both the space variability of rainfall and the (spatial) implementation of different nature-based solutions (NBS) on the hydrological response of an urban catchment southwest of Paris. This is done for three rainfall events using a physically-based, spatially distributed hydrological model, which was derived on a 10-m spatial and 3-min temporal resolution. The authors clearly show the impact of taking into account, or not, the spatial variability of rainfall. To a lesser extent, they found this for the spatial implementation of the NBS too. The authors then touch upon the effect of the combination of both effects, which may lead to insightful informa-

tion for urban water managers. I found this a relevant urban test case, especially the focus on the added value of a high-resolution, fully-distributed focus in urban hydrology, and I have read it with great interest. From that perspective, I think this manuscript may be considered for publication in HESS. I have, however, also some questions and suggestions to further improve the manuscript. Some are more major than others, but I hope that it helps improving the manuscript.

Below I describe some general comments, followed by more detailed comments line by line.

**2 General comments**

**2.1 Title**

The title gives the impression that the space variability of the hydrological response is studied, while the authors have studied the effect of the space variability of rainfall and NBS on the hydrological response. In addition, the authors present a very interesting case study, so perhaps it is good to mention this in title too (e.g. case study for Guyancourt).

**2.2 Space-time resolution of the hydrological model**

The authors state that the model has a spatial resolution of 10 m, but a DEM with a resolution of 25 m was used. Could the authors explain how this coarser DEM was used to derive the model topography on a 10 m spatial resolution? Moreover, the model was run with a temporal resolution of 3 min, whereas the temporal resolution of the rainfall data for 3.4 minutes. Could the authors indicate how they have downscaled the temporal resolution of the rainfall data? Hydrological model parametrization and

reproducibility of the NBS implementations The used hydrological model Multi-Hydro is shortly introduced in Section 2.3. In Section 3.1, the authors introduce the implementation of the different land use classes and the model parameterization. Although the authors do not have to introduce the entire model (the given references suffice), I would recommend more elaborately introducing the used model parameters and the effect they have, including the different land use classes, on the model. Hence, what can we expect from the given parameterization and land uses classes (i.e, differences in evapotranspiration, interception, storage capacity, etc.)? The same holds for the NBS scenarios in the model and their parameterization (for the green roofs, this was already done by the authors). For the NBS scenarios, also describe how they are implemented and what their expected storage increase, or effect, is. Why are the NBS scenarios applied in the way they are applied (in space, but also the density of the application)? Is this a feasible application or have you chosen this purely as a synthetic experiment? A more elaborate description would highly increase the reproducibility of this study, also with other hydrological models.

**2.3 Experimental setup**

I think the authors chose for an elegant application by using the multifractal approach. The authors write: "For instance, both 'hot spots' (extremes) of the rainfall and NBS are scarce and therefore could rarely coincide, i.e., rainfall spikes may fall more often elsewhere than on NBS. Similar questions can occur for less extreme events. The effective NBS performance could be therefore biased with respect to their potential performance due to this problem of intersection between rainfall intensity and NBS". This make me wonder if a synthetic experiment would have worked too. In such an experiment, you can exactly define where, how much and when the rainfall falls and the same for the location and density of the NBS applications. This would make it possible to do a larger sample analysis of the effects (the authors point on the need for this in the conclusion). Could the authors comment on this and if they agree that

something like that – or another approach – is possible, write something about it in the discussion?

**2.4   Verification metrics**

In the methods and results, the percent error is used. However, the authors compare different scenarios and approaches with each other. The percent error is generally used to compare a result with the true value or at least the theoretical value. It gives the feeling of an error, whereas you are not sure which one is right or wrong. Hence, using the percent difference throughout the manuscript may be a better choice.

**2.5   The results in a larger perspective**

In the last section of the results and the conclusions, I miss a bit the significance of the results. How does this relate to other literature and are the intersection results of importance when compared to the large effect of uniform or non-uniform rainfall inputs? In addition, what can the authors say about the effects of the chosen events on the results? This is of course a case study, but I would encourage the authors to put the results a bit into perspective.

**3   Specific comments**

Page 1, line 13: "of their hydrological responses sensitively depends". Do you mean that the hydrological model results depend on the mentioned two processes?

Page 1, line 24: It is not more pronounced, but there is a somewhat significant effect for the two NBS scenarios the authors mention.

Page 1, lines 26 – 27: What kind of flooding do you mean (i.e., flash, coastal, urban, fluvial and/or pluvial floods)? In case of just urban flooding, the sentence is fine like this. Otherwise, there are more essential drivers, such as land subsidence in deltas, deforestation, etc. (add some extra references in that case). Nevertheless, I would suggest to change "The increasing of extreme flooding risks" into "The increase of extreme flood risk" (possibly with an indication of what kind of flood risk).

Page 2, line 37: "parallel concepts". Perhaps I am not familiar enough with the concepts, but what kind of parallel concepts are meant here?

Page 2, line 44 – 45: For completeness, also write why the fully-distributed models are rarely used.

Page 2, line 49: What do you mean by complicated urban catchments?

Page 2, line 54: "Indeed, such models should better assess the hydrological performances of NBS on a smaller scale." Although I agree with the need for such models, be careful with this statement. It stands or falls with the presence of reliable high-resolution model forcing and parameterization (among others).

Page 2, line 55: "lack of high-resolution rainfall data". What is the desired resolution the authors are looking for? I.e., are rainfall estimates from X-band radars, commercial microwave links or personal weather stations a solution? I am aware of the challenge of (reliable, high-resolution) rainfall estimation in urban areas, but be a bit more specific about what the limitation is.

Page 2, lines 59 – 61: This is the first time you introduce the intersection between the spatial variability of rainfall and the NBS. Although I do understand what you mean, I would suggest devoting one sentence here to explain what is meant by that. In addition, what about the time variability of rainfall. What can you say about the importance of that?

Page 3, line 68: "Two different types of rainfall data". The data source is the same, so

I would rather suggest calling it two different rainfall processing approaches (gridded and catchment-averaged).

Page 4, lines 104 – 105: What is the rainfall amount that corresponds to this return period? That would make it easier to relate the studied events to the drainage capacity.

Page 4, lines 113 – 115: Could you tell a little bit more about how this tool works?

Page 8, lines 245 – 251: This is a clear explanation, which would be even more valuable when placed at the start of the section. Page 11, lines 317 – 324: It would be good to place a reference to Nash & Sutcliffe (1970) here.

Page 11, lines 328 – 338: The authors already describe some model results here. This fits better in the results, e.g. as first subsection. Side note, the model indeed performs well for the given study area.

Page 11, lines 339 – 340: Although I do agree with this conclusion, can the authors say something about the model performance (regarding simulated fluxes and/or states) on the grid level or at the sub-catchment scale? On the used high spatial resolution and in an urban setting, I know this is challenging. However, it would further support your conclusions, especially because you are focussing on spatial variability in the results.

Page 12, lines 361 – 363: This is exactly what you expect for the baseline situation. The difference in total runoff volume should not be too different, because the total rainfall volume should be the same for the gridded and uniform rainfall inputs. The small differences are an effect of differences on the grid scale (storage capacity, evapotranspiration, etc.), which are differently modelled when the input is uniform or non-uniform.

Page 14, lines 433 – 437: I think this is one of the most interesting (and important) results of the study. However, it is not very easy for the reader to make the comparison based on the figures (e.g. figures 14 and 16). Could the authors add a subplot to figure 16 indicating the differences between the two scenarios (so the difference between uniform and non-uniform rainfall plus the difference between the scenarios) and tell

somewhat more about it?

Page 15, conclusions: The abbreviations used throughout the text, are also directly used in the conclusions. If you read the entire text, this is clear, but for readers who quickly skim through the abstract and conclusions, I would suggest to (re-)introduce the meaning of these abbreviations.

Page 16, lines 479 – 480: "However, the RG scenarios appear to be less affected by the intersection effects, with a difference lower than 3% on peak flow and lower than 1 % on total runoff volume." This is indeed supported by the results, but in the results, you also discuss the reason for this small effect on the peak flow and runoff volume. It would be good to include that here too.

Page 16, lines 481 – 485: I fully agree with the authors that this hints towards using fully distributed hydrological models over semi-distributed or lumped models, but that is not exactly what is shown in the results. The authors do not benchmark the result with semi-distributed or lumped models, but rather focus on the rainfall variability and NBS variability on the discharge response. I would like to ask the authors to rephrase this paragraph a bit. Figures overall – Make sure the font size is readable and approximately the same font size is used for all the figures in the manuscript.

**4  Technical corrections**

Page 1, lines 17 – 19: For readability, I would suggest making two separate sentences out of this one.

Page 1, line 21: ", which is more pronounced than those of the total runoff volume." Do you mean, ", which is a stronger effect than the effect on the total runoff volume."?

Page 2, line 30: "results in rainfall transfer into runoff rapidly" becomes "result in a rapid transfer of rainfall into runoff".

Page 2, line 31: "The approach of expanded and upgraded the capacity of the existing drainage system" becomes "Expanding and upgrading the capacity of the existing drainage system".

Page 3, line 62: "such mentioned" becomes "the mentioned" and "over higher spatial resolutions" becomes "for higher spatial model resolutions".

Page 4, line 109: "a clear tendency towards growing number of somewhat shorter, but much heavier rainfall events, was perceived for this region" Suggested change: "A clear tendency towards a growing number of shorter duration, but higher intensity rainfall events is perceived for this region".

Page 6, line 184: Remove the ')' before the end of the sentence.

Page 7, line 201: "simulated under both different types of rainfall" becomes "simulated for both rainfall inputs".

Page 18 and 19, lines 559 – 567: The references are not alphabetical here. Perhaps also at other lines.

Figure 2: I may be wrong, but it seems that the legend colour for forest is not exactly the same as the colour used in the map. Figure 3: The rainbow colour map is not always intuitive, also here with respect to the rainfall amounts. I recommend using a more intuitive colour map. Some explanations and inspiration can be found in Crameri et al. (2020).

Figure 10: The authors refer in the caption to (a) EV1, (b) EV2, (c) EV3, but the letters (a) – (c) are not shown in the figure.

Figure 12c: It would be better to show the ratio on a logarithmic axis.

Figure 14: I spotted a minor typo in the figure title (Percenatge instead of percentage).

[Figure]

**5  References**

Crameri, F., Shephard, G.E.  Heron, P.J (2020).  The misuse of colour in science communication.  Nature Communications, 11, 5444 (2020).  https://doi.org/10.1038/s41467-020-19160-7.

Nash, J. E.,  Sutcliffe, J. V. (1970).  River flow forecasting through conceptual models part I - A discussion of principles.  Journal of Hydrology, 10(3), 282–290.  https://doi.org/10.1016/0022-1694(70)90255-6.

---

## Author Comment (AC1) · 23 Feb 2021

**Dear referees,**

**We would like to thank you for your careful reviews and constructive suggestions with regard to our manuscript. Your thoughtful remarks and suggestions enabled us to enrich the paper and bring complementary information and details that make it more straightforward. We therefore took into account all your comments, queries and suggestions.**

**We also thank you for the clarity of your reports that made the correction work easier, as well as our point-by-point responses that follow. Most of the time, our replies are presented under the form of modified texts ready to be inserted to obtain a revised version, which will also take into account the editor's comments and suggestions. Referees' comments appear in italics and black colour, our reply in normal font and blue colour, and corresponding modifications proposed for the text in the manuscript are in underline font and red colour.**

**Responses to Referee 1**

*Referee comments 1 :*

*In this article, the authors aim to investigate the uncertainty of hydrological responses in various NBS scenarios resulting from the spatial variability of rainfall and the heterogeneous distribution of NBS at the urban catchment scale. I find the manuscript to be quite suitable for HESS and presents a straight forward method which is intended to provide means to modelling works for NBS at the urban scale.*

*Overall, the manuscript is well presented. I would like to suggest a Minor revision for this paper, but there are a few points that should be addressed to improve its quality.*

*1. You use 25-m resolution DEM but the model was implemented with a 10 m spatial resolution, could you please make a comment or discussion about this? Moreover, 25-m resolution DEM seems relatively quite rough to use for an urban area. Do you think this is the limitation of your study? If you use a more detailed DEM, will it impact your result?*

**Author's answer:**
Thanks to the referee's suggestions, the authors added the following underlined sentences (at lines 193-194 of the previous version) to enhance the comprehension and readability of the manuscript:

"Besides the land use, the elevation is also assigned to each pixel of the model. For this purpose, the interpolation was used to downscale the raw DEM data from 25 m to 10 m (DEM25-10) to incorporate it with the model resolution. More precisely, each pixel was first subdivided into 25 equal sub-pixels as a proxy of the 5 m resolution, then the elevation data were up-scaled 4 by 4 pixels to produce the 10 m interpolation of the original elevation.
While the 25 m resolution DEM may seem too coarse to use for an urban area, it did not limit the study in any way because the catchment is relatively flat. To test this, we up-scaled the raw 5 m DEM data to adapt them to the model resolution (DEM5-10). Table. 1A presents the results of the statistical analysis of DEM25-10 and DEM5-10, which are so similar that the difference could not impact the results. For instance, the Root Mean Square Error is about 0.26, and the correlation coefficient is around 0.99. Besides, the ensemble of the data actually available for the Guyancourt watershed would need to be more detailed to make it worth going to a higher resolution of the model.

| Statistic metrics | DEM25-10 | DEM5-10 |
|---|---|---|
| Median | 143.3 | 143.4 |
| Mean | 160.1 | 160.1 |
| Maximum | 175.4 | 175.9 |
| Minimum | 143.0 | 143.3 |
| Standard deviation | 80.2 | 80.2 |
| Root Mean Square Error | 0.26 | |
| Correlation coefficient | 0.99 | |
| Maximum difference | 5.3 | |
| Mean difference | 0.01 | |

**Table 1A . The statistical comparison of DEM5-10 and DEM25-10. This table plans to be inserted in the revised manuscript after the Table 2.**

*Referee comments 2:*

*2. Figure 5, In the legend it should include an abbreviation of each NBS measures e.g., Porous pavement (PP), Rain garden (RG),. . .*

**Author's answer:**

Thanks to the referee's suggestion, we have revised the Figure 5 including the abbreviation of each NBS measures.

[Figure]

**Figure 5. NBS scenarios including PP1, PP2, RG1, RG2, GR1, GR2, Combined1, and Combined2. The rectangular area presented in the PP1 scenario is the example area for applying fractal analysis.**

*3. Please consider the results of validation from Line 328 to the Results section as this is results of the validation of your baseline scenario.*

**Author's answer:**

Thanks to your suggestion. We have moved the results of validation (from line 328 to line 340) to the Results section and added some discussions in the revised version of the manuscript.

**4 Results and discussion**

**4.1 Validation of baseline scenario**

Regarding the water levels observed and those simulated in the baseline scenario, the model indeed performs well for the studied area. The *NSE* coefficients and the *PE* indicators validated Multi-Hydro's performance (see Table 4). Indeed, for the three distributed rainfall events (Figure 9), the *NSE* are larger than 0.9, and *PE* are lower than 5 %. For the uniform rainfall event of EV2, the model represents the water levels with *NSE* equal to 0.95, and *PE* equal to 1.96 %: only a slight overestimation of the observed water levels is observed between hours 4 to 7. For the uniform rainfall of EV1 and EV3, the temporal evolutions of simulated water levels slightly underestimate the observed ones, with *NSE* around 0.8, as well as *PE* around 7 %. Regarding the temporal evolutions of simulated water levels under the distributed rainfall of EV1 and EV3, they are more consistent with the observed ones. The reason is that the rainfall intensities of the distributed rainfall are generally higher than those of the uniform rainfall at the storage basin location. Namely, in uniform rainfall events, the accumulated water levels in the storage basin are less than that of in distributed rainfall events. Overall, the distributed rainfall gives slightly better results, and the simulated water levels using uniform rainfall also match sufficiently well the observed ones to validate the Multi-Hydro implementation in the Guyancourt catchment.

Regarding the validation results, the scalability of Multi-Hydro allowed us to define the optimal resolution to finely reproduce the spatial heterogeneity of the watershed. Remember that this resolution is the ratio between the external scale of the watershed and the scale of the grid. The heterogeneity mentioned above propagates from the smallest scale to the largest, impacting the simulation results in any through the hierarchy of spatial scales of the watershed. It should be understood that the selected 10 m grid scale is not the smallest scale possible, but the optimal one to ensure a good balance between, for example, sufficient heterogeneity and the required quantity of the data required, again in precision valuable and involved computing time involved. As discussed in Section 3.2, the spatial heterogeneity for each of the NBS scenarios evolves with the fractal dimension on two scale ranges: the asset implementation scales (10m - 80m) and the larger basin scales. Such an evolution remains fully compatible with the intrinsic scalability of Multi-Hydro, which makes it particularly suitable and sufficiently reliable to study the impacts of the spatial variability of hydrological responses in different NBS scenarios.

*4. For Figure 12 and 16, can you discuss more on why percentage errors of peak flow are much higher than total runoff volume.*

**Author's answer:**

Thanks to the referee's suggestion. The authors added the discussions for Figures 12 and 16 to enhance the

comprehension and readability of the manuscript (in our revised manuscript, the verification metrics of percentage error was changed to percentage difference as the metrics throughout the manuscript. The corresponding abbreviation for percentage difference on peak flow was modified to '$PD_{Qp}$', and the percentage difference on total runoff volume was modified to '$PD_V$').

As shown in Figure 12 and Figure 16, the percentage difference on peak flow is much higher than that of the total runoff for each scenario. These results can be explained by the fact that the spatial variability of rainfall intensity at the largest rainfall peak is strong in all three rainfall events, while the total rainfall volume for the gridded and uniform rainfall inputs is the same. Being an integrative variable of the rainfall over the watershed, the total volume of runoff should not differ much for gridded and uniform rainfall inputs. This small $PD_V$ is influenced by the differences on the grid scale (storage capacity, infiltration, etc.), which are differently modelled when the input is uniform or non-uniform. For these three rainfall events, the spatial variability of total rainfall depth is less pronounced with respect to the spatial variability of the rainfall intensity at the largest rainfall peak, and also there is no highly localized storm cell. Figure 3 (top) displays the rainfall intensity at the largest rainfall peak (per radar pixel) over the Guyancourt catchment area for the three studied rainfall events. It is noticed that the highest rainfall peak of the gridded rainfall is very variable in space, which enlarged the discrepancy with the corresponding uniform rainfall data, resulting in a significant impact on the peak flow of each NBS scenario that simulated with two different rainfall data. However, the cumulative rainfall of gridded rainfall data is not very variable in space (see Figure 3 middle). For instance, the standard deviation (*SD*) of the cumulative rainfall of the three rainfall events is around 1 mm, which indicates that the spatial variability of the distributed rainfall is not very pronounced at most of the time steps. Thus, the difference between gridded rainfall data and uniform rainfall data is relatively small during the whole rainfall period. Finally, the simulated flow of NBS scenarios under two different rainfall inputs is similar in most time steps, resulting in the percentage difference on the total runoff volume of NBS scenarios (simulated by gridded rainfall input and uniform rainfall input) is not significant.

[Figure]

**Figure 3: Top: The rainfall intensity at the largest rainfall peak (per radar pixel) over the Guyancourt catchment area for the three studied rainfall events. Middle: Cumulative rainfall depths (per radar pixel) over the Guyancourt catchment area for the three studied rainfall events. Bottom: Time evolution of rainfall rate (mm h⁻¹) and cumulative rainfall (mm) of the three uniform rainfall events over the whole catchment.**

[Figure]

**Figure 12. (a) Percentage difference on peak flow of the baseline scenario and the first set of NBS scenarios under the three distributed rainfall events and the three uniform rainfall events. (b) Percentage difference on total runoff volume of the baseline scenario and the first set of NBS scenarios under the three distributed rainfall events and the three uniform rainfall events. (c) The ratio of peak flow between the scenarios under the distributed rainfall and the scenarios under the uniform rainfall.**

[Figure]

**Figure 16. (a) Percentage difference on peak flow of all NBS scenarios under the three distributed rainfall events and the three uniform rainfall events. (b) Percentage difference on total runoff volume of all NBS scenarios under the three distributed rainfall events and the three uniform rainfall events. (c) Difference of $PD_{Qp}$ between the same types of NBS scenario. (d) Difference of $PD_V$ between the same types of NBS scenario.**

---

## Author Response (AR2)

**Point-by-point reply, manuscript hess-2020-468**

Dear editor and referees,

We would like to thank you for your careful reviews and constructive suggestions with regard to our manuscript. Your thoughtful remarks and suggestions enabled us to enrich the paper and bring complementary information and details that make it more straightforward. We therefore took into account all your comments, queries and suggestions.

We also thank you for the clarity of your reports that made the correction work easier, as well as our point-by-point responses that follow. Most of the time, our replies are presented under the form of modified texts that have been inserted into the revised version. Editor and referees' comments appear in italics and black colour, our reply in normal font and blue colour, and corresponding modifications proposed for the text in the manuscript are in underline font and red colour. For the sake of convenience, we produced the marked-up manuscript in PDF, and all the modifications can be traced in the text (the new text in the underline font and blue colour, the deleted in  characters).

Best regards,
Yangzi QIU

**Editor Decision: Publish subject to revisions (further review by editor and referees)**

***Comments to the Author:***

*The authors analyze the impact of the spatial variability of rainfall and of the spatial arrangement of NBS on hydrological outputs of NBS scenarios at the urban catchment scale.*

*The manuscript has been revised by two reviewers who suggested (overall) moderate revisions, mainly related to the inclusion in the manuscript of additional relevant details (such as the space-time resolution, model parameters, model performance,…).*

*The authors provided a detailed answer to the reviewers' comments, highlighting the main changes they envision in the manuscript.*

*My main concern regards the application of the Surface-Subsurface Model (Multi-Hydro model). In their analysis the authors did not model the infiltration process, in order to simply the procedure. Although I understand the authors' attempt to simplify the problem, the infiltration process can be extremely important for the problem considered and cannot be neglected. One aspect that the authors should consider during the revision of their work is to further elucidate this aspect.*

We thank Editor Monica Riva for constructive suggestions concerning our manuscript and for this chance to improve our manuscript. We have carefully considered and addressed all your comments, as detailed in the following pages.

*Author's answer:*
Thanks to editor's suggestion. The authors revised the Section 2.3 Multi-Hydro model (at line 194 and lines 204-221 of the revised version) to enhance the comprehension and readability of the manuscript:

**2.3 Multi-Hydro model**

Multi-Hydro constitutes the interactive core among the four open-source modules (rainfall, surface, groundwater, drainage) that represent essential elements of the hydrological cycle in urban environment.

The surface module (MHSC) of Multi-Hydro uses the code of the Two-Dimensional Runoff Erosion and Export (TREX) model that computes the interception, storage and infiltration occurring at each pixel in terms of the properties of each land use (Velleux et al., 2008). The infiltration process of the surface module is governed by simplification of Green and Ampt equation (see p. 4 of the TREX user manual). The diffusive wave approximation of the Saint-Venant equations is used for calculating the overland flow, following the conservation of mass and momentum equations.

The groundwater recharge and solute transport (Riva et al., 2006; Mooers et al., 2018) are the other significant aspects of the hydrological cycle. The groundwater module (MHGC) is based on the Variably Saturated and 2-Dimensional Transport (VS2DT) model developed by the U.S. Geological Survey. This module can be used to simulate variably saturated transient water flow and solute transport in one or two dimensions (Lappala et al., 1987; Healy, 1990).

In this study, we used the Multi-Hydro interaction between the surface module and the drainage module to focus on the rainfall-runoff modelling of NBS scenarios. In urban areas, groundwater can produce infiltration into the drainage pipes due to cracks in the structure (see Lucas and Sample, 2015 for an example). The absence of long recession limb on the hydrographs indicates there is no such problems on the studied watershed. Groundwater can also eventually contribute to surface flooding when it is saturated. Such phenomenon did not occur on the studied area due to its pedology and the considered (not extreme) rainfall events. For these reasons, groundwater (as evapotranspiration) has not been considered in this study. That has been focused on the fast response of the watershed at the rainfall event scale.

**Responses to Anonymous Referee #1**

*In this article, the authors aim to investigate the uncertainty of hydrological responses in various NBS scenarios resulting from the spatial variability of rainfall and the heterogeneous distribution of NBS at the urban catchment scale. I find the manuscript to be quite suitable for HESS and presents a straight forward method which is intended to provide means to modelling works for NBS at the urban scale.*
*Overall, the manuscript is well presented. I would like to suggest a Minor revision for this paper, but there are a few points that should be addressed to improve its quality.*

We thank Anonymous Referee #1 for the time she/he spent on our manuscript and for the useful and constructive comments that will help to improve the quality of the manuscript. We have carefully considered and addressed all her/his comments in the following.

*Referee comments 1:*
*1. You use 25-m resolution DEM but the model was implemented with a 10 m spatial resolution, could you please make a comment or discussion about this? Moreover, 25-m resolution DEM seems relatively quite rough to use for an urban area. Do you think this is the limitation of your study? If you use a more detailed DEM, will it impact your result?*

**Author's answer:**
Thanks to the referee's suggestions, the authors added the following underlined sentences (at lines 253-263 of the revised version) to enhance the comprehension and readability of the manuscript:

"Besides the land use, the elevation is also assigned to each pixel of the model. For this purpose, the interpolation was used to downscale the raw DEM data from 25 m to 10 m (DEM25-10) to incorporate it with the model resolution. More precisely, each pixel was first subdivided into 25 equal sub-pixels as a proxy of the 5 m resolution, then the elevation data were up-scaled 4 by 4 pixels to produce the 10 m interpolation of the original elevation.
While the 25 m resolution DEM may seem too coarse to use for an urban area, it did not limit the study in any way because the catchment is relatively flat. To test this, we up-scaled the raw 5 m DEM data to adapt them to the model resolution (DEM5-10). Table. 3 presents the results of the statistical analysis of DEM25-10 and DEM5-10, which are so similar that the difference could not impact the results. For instance, the Root Mean Square Error is about 0.26, and the correlation coefficient is around 0.99. Besides, the ensemble of the data actually available for the Guyancourt watershed would need to be more detailed to make it worth going to a higher resolution of the model.

| Statistic metrics | DEM25-10 | DEM5-10 |
|---|---|---|
| Median | 143.3 | 143.4 |
| Mean | 160.1 | 160.1 |
| Maximum | 175.4 | 175.9 |
| Minimum | 143.0 | 143.3 |
| Standard deviation | 80.2 | 80.2 |
| Root Mean Square Error | 0.26 | |
| Correlation coefficient | 0.99 | |
| Maximum difference | 5.3 | |
| Mean difference | 0.01 | |

**Table 3. The statistical comparison of DEM5-10 and DEM25-10.**

*Referee comments 2:*

*2. Figure 5, In the legend it should include an abbreviation of each NBS measures e.g., Porous pavement (PP), Rain garden (RG),. . .*

**Author's answer:**

Thanks to the referee's suggestion, we have revised the Figure 5 (in the revised manuscript is Figure 6) including the abbreviation of each NBS measures.

[Figure]

**Figure 5. Two scenarios for each of NBS implementation, including porous pavement (PP1, PP2), rain garden (RG1, RG2), green roof (GR1, GR2), Combined1 and Combined2. The rectangular area presented in the PP1 scenario is the example area for applying fractal analysis.**

*Referee comments 3:*

*3. Please consider the results of validation from Line 328 to the Results section as this is results of the validation of your baseline scenario.*

Author's answer:
Thanks to your suggestion. We have moved the results of validation to the Section. 4.1 (revised version) and added some discussions in the revised version of the manuscript (at lines 428-449).

**4 Results and discussion**

**4.1 Validation of baseline scenario**

Regarding the water levels observed and those simulated in the baseline scenario, the model indeed performs well for the studied area. The *NSE* coefficients and the *PE* indicators validated Multi-Hydro's performance (see Table 4). Indeed, for the three distributed rainfall events (Figure 9), the *NSE* are larger than 0.9, and *PE* are lower than 5 %. For the uniform rainfall event of EV2, the model represents the water levels with *NSE* equal to 0.95, and *PE* equal to 1.96 %: only a slight overestimation of the observed water levels is observed between hours 4 to 7. For the uniform rainfall of EV1 and EV3, the temporal evolutions of simulated water levels slightly underestimate the observed ones, with *NSE* around 0.8, as well as *PE* around 7 %. Regarding the temporal evolutions of simulated water levels under the distributed rainfall of EV1 and EV3, they are more consistent with the observed ones. The reason is that the rainfall intensities of the distributed rainfall are generally higher than those of the uniform rainfall at the storage basin location. Namely, in uniform rainfall events, the accumulated water levels in the storage basin are less than that of in distributed rainfall events. Overall, the distributed rainfall gives slightly better results, and the simulated water levels using uniform rainfall also match sufficiently well the observed ones to validate the Multi-Hydro implementation in the Guyancourt catchment.

Regarding the validation results, the scalability of Multi-Hydro allowed us to define the optimal resolution to finely reproduce the spatial heterogeneity of the watershed. Remember that this resolution is the ratio between the external scale of the watershed and the scale of the grid. The heterogeneity mentioned above propagates from the smallest scale to the largest, impacting the simulation results in any through the hierarchy of spatial scales of the watershed. It should be understood that the selected 10 m grid scale is not the smallest scale possible, but the optimal one to ensure a good balance between, for example, sufficient heterogeneity and the required quantity of the data required, again in precision valuable and involved computing time involved. As discussed in Section 3.2, the spatial heterogeneity for each of the NBS scenarios evolves with the fractal dimension on two scale ranges: the asset implementation scales (10m - 80m) and the larger basin scales. Such an evolution remains fully compatible with the intrinsic scalability of Multi-Hydro, which makes it particularly suitable and sufficiently reliable to study the impacts of the spatial variability of hydrological responses in different NBS scenarios.

***Referee comments 4:***
*4. For Figure 12 and 16, can you discuss more on why percentage errors of peak flow are much higher than total runoff volume.*

Author's answer:
Thanks to the referee's suggestion. The authors added the discussions for Figures 12 and 16 (in revised manuscript is Figures 13 and 17) to enhance the comprehension and readability of the manuscript (in our revised manuscript, the verification metrics of percentage error was changed to percentage difference as the metrics throughout the manuscript. The corresponding abbreviation for percentage difference on peak flow was modified to '$PD_{Qp}$', and the percentage difference on total runoff volume was modified to '$PD_V$').

As shown in Figure 12 and Figure 16, the percentage difference on peak flow is much higher than that of the total

runoff for each scenario. These results can be explained by the fact that the spatial variability of rainfall intensity at the largest rainfall peak is strong in all three rainfall events, while the total rainfall volume for the gridded and uniform rainfall inputs is the same. This small $PD_V$ is influenced by the differences on the grid scale (storage capacity, infiltration, etc.), which are differently modelled when the input is uniform or non-uniform. For these three rainfall events, the spatial variability of total rainfall depth is less pronounced with respect to the spatial variability of the rainfall intensity at the largest rainfall peak, and also there is no highly localized storm cell. Figure 3 (top) displays the rainfall intensity at the largest rainfall peak (per radar pixel) over the Guyancourt catchment area for the three studied rainfall events. It is noticed that the highest rainfall peak of the distributed rainfall is very variable in space, which enlarged the discrepancy with the corresponding uniform rainfall, resulting in a significant impact on the peak flow of each NBS scenario that simulated with two different rainfall inputs. However, the cumulative rainfall of the distributed rainfall input is not very variable in space (see Figure 3 middle). For instance, the standard deviation (SD) of the cumulative rainfall of the three rainfall events is around 1 mm, which indicates that the spatial variability of the distributed rainfall is not very pronounced at most of the time steps. Thus, the difference between distributed rainfall and uniform rainfall is relatively small during the whole rainfall period. Finally, the simulated flow of NBS scenarios under two different rainfall inputs is similar in most time steps, resulting in the percentage difference on the total runoff volume of NBS scenarios (simulated by distributed rainfall and uniform rainfall) is not significant.

[Figure]

**Figure 3: Top: The rainfall intensity at the largest rainfall peak (per radar pixel) over the Guyancourt catchment area for the three studied rainfall events; middle: cumulative rainfall depths (per radar pixel) over the Guyancourt catchment area for the three studied rainfall events; bottom: time evolution of rainfall rate (mm h$^{-1}$) and cumulative rainfall (mm) of the three uniform rainfall events over the whole catchment.**

[Figure]

**Figure 12. (a) The ratio of peak flow between the scenarios under the distributed rainfall and the scenarios under the uniform rainfall; (b) percentage difference on peak flow of the baseline scenario and the first set of NBS scenarios under the three distributed rainfall events and the three uniform rainfall events; (c) percentage difference on total runoff volume of the baseline scenario and the first set of NBS scenarios under the three distributed rainfall events and the three uniform rainfall events.**

[Figure]

**Figure 16. (a) Percentage difference on peak flow of all NBS scenarios under the three distributed rainfall events and the three uniform rainfall events; (b) percentage difference on total runoff volume of all NBS scenarios under the three distributed rainfall events and the three uniform rainfall events. (c) Difference of $PD_{Qp}$ between the same types of NBS scenario. (d) Difference of $PD_V$ between the same types of NBS scenario.**

**Response to Referee Ruben Imhoff**
ruben.imhoff@deltares.nl

*1 Summary*

*The manuscript describes the effect of both the space variability of rainfall and the (spatial) implementation of different nature-based solutions (NBS) on the hydrological response of an urban catchment southwest of Paris. This is done for three rainfall events using a physically-based, spatially distributed hydrological model, which was derived on a 10-m spatial and 3-min temporal resolution. The authors clearly show the impact of taking into account, or not, the spatial variability of rainfall. To a lesser extent, they found this for the spatial implementation of the NBS too. The authors then touch upon the effect of the combination of both effects, which may lead to insightful information for urban water managers. I found this a relevant urban test case, especially the focus on the added value of a high-resolution, fully-distributed focus in urban hydrology, and I have read it with great interest. From that perspective, I think this manuscript may be considered for publication in HESS. I have, however, also some questions and suggestions to further improve the manuscript. Some are more major than others, but I hope that it helps improving the manuscript.*

*Below I describe some general comments, followed by more detailed comments line by line.*

We thank Referee Ruben Imhoff for the time he spent on our manuscript and for the useful and constructive comments that will help to improve the quality of the manuscript. We have carefully considered and addressed all his comments in the following.

**Referee general comments:**

**Referee comments 2.1 Title**

*The title gives the impression that the space variability of the hydrological response is studied, while the authors have studied the effect of the space variability of rainfall and NBS on the hydrological response. In addition, the authors present a very interesting case study, so perhaps it is good to mention this in title too (e.g. case study for Guyancourt).*

**Author's answer:**

As suggested by referee, we changed the title to 'Space variability impacts on hydrological responses of Nature-Based Solutions and the resulting uncertainty: a case study of Guyancourt (France)'

**Referee comments 2.2 Space-time resolution of the hydrological model**

*The authors state that the model has a spatial resolution of 10 m, but a DEM with a resolution of 25 m was used. Could the authors explain how this coarser DEM was used to derive the model topography on a 10 m spatial resolution?*

**Author's answer:**

Thanks to the referee's suggestions, the authors added the following underlined sentences (at lines 253-263 of the revised version) to enhance the comprehension and readability of the manuscript:

"Besides the land use, the elevation is also assigned to each pixel of the model. For this purpose, the interpolation was used to downscale the raw DEM data from 25 m to 10 m (DEM25-10) to incorporate it with the model resolution.

More precisely, each pixel was first subdivided into 25 equal sub-pixels as a proxy of the 5 m resolution, then the elevation data were up-scaled 4 by 4 pixels to produce the 10 m interpolation of the original elevation.

While the 25 m resolution DEM may seem too coarse to use for an urban area, it did not limit the study in any way because the catchment is relatively flat. To test this, we up-scaled the raw 5 m DEM data to adapt them to the model resolution (DEM5-10). Table. 3 presents the results of the statistical analysis of DEM25-10 and DEM5-10, which are so similar that the difference could not impact the results. For instance, the Root Mean Square Error is about 0.26, and the correlation coefficient is around 0.99. Besides, the ensemble of the data actually available for the Guyancourt watershed would need to be more detailed to make it worth going to a higher resolution of the model.

| Statistic metrics | DEM25-10 | DEM5-10 |
|---|---|---|
| Median | 143.3 | 143.4 |
| Mean | 160.1 | 160.1 |
| Maximum | 175.4 | 175.9 |
| Minimum | 143.0 | 143.3 |
| Standard deviation | 80.2 | 80.2 |
| Root Mean Square Error | 0.26 | |
| Correlation coefficient | 0.99 | |
| Maximum difference | 5.3 | |
| Mean difference | 0.01 | |

**Table 3. The statistical comparison of DEM5-10 and DEM25-10.**

*Moreover, the model was run with a temporal resolution of 3 min, whereas the temporal resolution of the rainfall data for 3.4 minutes. Could the authors indicate how they have downscaled the temporal resolution of the rainfall data?*

**Author's answer:**
The authors would like to thank the reviewer once more for this constructive comment. We added the following underlined sentences (at lines 231-237 of the revised version) to enhance the comprehension and readability of the manuscript:

Similar to the adaptation of spatial resolution of rainfall input to that of the model, the rainfall input for Multi-Hydro has been also time interpolated from the X-band radar measurements, as follows:

$$R_m(j) = \frac{\sum_i R_r(i)|\Delta_m(j) \cap \Delta_r(i)|}{\delta_m}$$

where $R_m(j)$ is the rainfall rate during the j-th time interval $\Delta_m(j)$ of the model, $R_r(i)$ is the rainfall rate during the i-th time interval $\Delta_r(i)$ of the X-band radar. $|\Delta|$ denotes the length of any interval $\Delta$ and $\delta_m = |\Delta_m|$ is the length of any time interval of the model. Note that while the duration of the time loop to generate the model outputs is 3 min (to keep it comparable with the X-band radar time interval), $\delta_m = 1$ minute for the rain input to Multi-Hydro.

*Hydrological model parametrization and reproducibility of the NBS implementations The used hydrological model Multi-Hydro is shortly introduced in Section 2.3. In Section 3.1, the authors introduce the implementation of the different land use classes and the model parameterization. Although the authors do not have to introduce the entire model (the given references suffice), I would recommend more elaborately introducing the used model parameters and the effect they have, including the different land use classes, on the model. Hence, what can we expect from the given parameterization and land uses classes (i.e, differences in evapotranspiration, interception, storage capacity, etc.)? The same holds for the NBS scenarios in the model and their parameterization (for the green roofs, this was already done by the authors).*

**Author's answer:**

Thanks to the referee's suggestion. The authors added the following underlined sentences (at lines 243-248, and lines 264-271 of the revised version) to the revised version of the manuscript:

For this study, all the standard model parameters related to the land use classification were selected from the Multi-Hydro manual (Giangola-Murzyn et al., 2014). The most important parameters are Manning's coefficient (no unit), hydraulic conductivity (m s$^{-1}$) and interception (mm), as they are shown in Table 2. As already indicated, the Multi-Hydro does not use the traditional calibration of these parameters. If their most common values are always used, the reliable heterogeneity of the watershed for each case study is obtained by a rapid optimization of the spatio-temporal resolution of the model, with possibly refined classes of the land use and their orders (Ichiba et al., 2018).

As the most considered NBS correspond to more specific land uses, they are characterised with different retention capacities, the related parameters are based on the literatures (Dussaillant et al., 2004; Kuang et al., 2011; Park et al., 2014). To be more specific, the rain gardens (RG) characterised with the depression depth of 0.3 m. Thus, the storage capacity of RG is about 300 L m$^{-2}$. For the porous pavements (PPs), the thickness of pavements is 0.21 m (i.e., pavement (0.08 m), bedding material (0.03 m) and base material (0.1 m)). The porosity of pavement, bedding material, and base material is 5.4 %, 28.29 % and 22.66 %, respectively. This indicates that the storage capacity of PP is approximately 74 L m$^{-2}$ in this study. For these two NBS measures, a simple procedure represents both infiltration and storage processes has been carried out. For each time step, if the rainfall rate lower than infiltration rate of porous pavement/rain garden, the water is stored. If not, then the ponding occurs.

**Table 2. Hydrological parameters for each land use class.**

| Land use | Manning's coefficient (no units) | Hydraulic conductivity (m s$^{-1}$) | Interception (mm) |
|---|---|---|---|
| Impervious surfaces (road, house, parking …) | 0.012 | 1.0e-10 | 1.9 |
| Gullies | 0.9 | 1.0e-0 | 0 |
| Grass | 0.15 | 1.9e-6 | 3.81 |
| Forest | 0.8 | 1.9e-6 | 7.62 |
| Water | 0.9 | 1.0e-0 | 100 |
| Porous pavement | 0.014 | 1.0e-4 | 2.14 |
| Rain garden | 0.2 | 1.9e-5 | 7.62 |
| Green roof | 0.14 | 3.3e-4 | 3.81 |

*For the NBS scenarios, also describe how they are implemented and what their expected storage increase, or effect, is. Why are the NBS scenarios applied in the way they are applied (in space, but also the density of the application)? Is this a feasible application or have you chosen this purely as a synthetic experiment? A more elaborate description would highly increase the reproducibility of this study, also with other hydrological models.*

**Author's answer:**

Thanks to the referee's suggestion. The authors added the following underlined sentences (at lines 283-297 of revised manuscript) to enhance the comprehension and readability of the manuscript:

For each scenario, the corresponding NBS are implemented heterogeneously over the catchment, while respecting the local catchment conditions and stormwater management requirements. For instance, with the help of the detailed land use GIS data, we initially selected all the buildings having flat roofs, then these impervious roofs were converted into green roofs for the GR1 scenarios by adapting the land use data.

The second set of NBS scenarios (PP2, RG2, GR2, and Combined2) was proposed with a different arrangement to assess the potential effects of a heterogeneous implementation of NBS at the urban catchment scale. For each pair of scenarios with a given type of NBSs, their implementation occupies the same percentage of the space over the whole catchment, but differs significantly in terms of spatial distributions of the considered asset. Considering now the roofs with certain slopes (≤ 15°), they can be also used to implement green roofs (Stanic et al., 2018). The impervious roofs that satisfied this condition were converted into small and light green roofs and used for the GR2 scenario. While the two scenarios (GR1 and GR2) occupy the same percentage of the whole catchment, their density is different, simply because of the difference of original densities of the buildings. The designing process for other NBS scenarios follows a somewhat similar logic. All details concerning the scenarios of the NBS implementations, including a detailed description of each NBS and the percentage of the space required for its implementation at 10 m resolution, are presented in Table 3, while the maps of the resulting land use are illustrated on Fig 5.

**Table 3. The details of simulation: NBS scenarios.**

| NBS measure | Scenario | Proportion of implementation in whole catchment / selected area (after rasterization) | $D_F$ of NBS in small scale/ large scale (after rasterization) | Description of scenario |
|---|---|---|---|---|
| Porous pavement (PP) | PP1 | 8.0 %/13.8 % | 1.14/1.92 | Porous pavements were implemented on the non-driveways (width equal and less than 2.5 m) and some parking lots. |
| | PP2 | 8.0 %/10.1 % | 1.21/1.79 | Porous pavements were implemented on secondary driveways (width between 2.5 m to 5 m). |
| Rain garden (RG) | RG1 | 8.2 %/6 % | 0.93/1.77 | The low elevation greenbelts around houses were implemented by rain gardens, which can collect and store up the surface runoff from surrounding impermeable areas before infiltration on site. When rain garden saturated, the redundant surface runoff will drain into the drainage system. |
| | RG2 | 8.2 %/7 % | 1.04/1.78 | The low elevation greenbelts around public buildings and parking lots. |
| Green roof (GR) | GR1 | 8.6 %/13.5 % | 1.18/1.87 | Small and light green roofs consisting of a soil layer and a storage layer are implemented on all flat roofs. |
| | GR2 | 8.6 %/6 % | 1.05/1.75 | Impervious roofs with slightly slope ($\leq$ 15°) were converted to small and light green roofs (Stanic et al., 2018). |
| NBS combinations | Combined1 | 24.8 %/38.5 % | 1.59/1.95 | A combination of PP1, RG1, GR1 |
| | Combined2 | 24.8 %/30.4 % | 1.45/1.98 | A combination of PP2, RG2, GR2 |

*Referee comments 2.3 Experimental setup*

*I think the authors chose for an elegant application by using the multifractal approach. The authors write: "For instance, both 'hot spots' (extremes) of the rainfall and NBS are scarce and therefore could rarely coincide, i.e., rainfall spikes may fall more often elsewhere than on NBS. Similar questions can occur for less extreme events. The effective NBS performance could be therefore biased with respect to their potential performance due to this problem of intersection between rainfall intensity and NBS". This make me wonder if a synthetic experiment would have worked too. In such an experiment, you can exactly define where, how much and when the rainfall falls and the same for the location and density of the NBS applications. This would make it possible to do a larger sample analysis of the effects (the authors point on the need for this in the conclusion). Could the authors comment on this and if they agree that something like that – or another approach – is possible, write something about it in the discussion?*

**Author's answer:**

Thanks to the referee's suggestion, we designed a synthetic experiment with the help of four synthetic rainfall events (EV4-EV7) and GR scenarios (GR1 and GR2). We added the description of the synthetic experiment and the corresponding results and discussions in our revised manuscript.

To deepen the knowledge of the intersection effects between the spatial variability of rainfall and spatial distribution of NBS, we also considered 4 synthetic rain events (EV4-EV7). All these events are based on the EV3U, by selecting the 2 hours period with the highest rainfall peak around 35 mm $h^{-1}$ (catchment-averaged), as illustrated on Figure 4a. However, during the 3 min that lasted the largest rainfall peak of the EV3U, a new space distribution and/or intensity of the rainfall was imposed for each synthetic rainfall event. As shown in Figure 4b, the catchment-averaged maximum rainfall peak is about 37 mm $h^{-1}$ for the EV4, and the corresponding catchment-averaged cumulative rainfall is about 4 mm. During these 3 minutes, the rainfall was binary re-distributed in space (Figure 4c), with the maximum intensity around 55 mm $h^{-1}$. For the remaining synthetic rain events, this binary distribution was modified as follows (see Figure 4d-f): the same maximum intensity of 55 mm $h^{-1}$ and zero rainfall elsewhere (EV5), the maximum intensity of 17 mm $h^{-1}$ and zero rainfall elsewhere (EV6), and the maximum intensity of 55 mm $h^{-1}$ has been replaced by zero rainfall (EV7).

For this experiment, the GR1 and GR2 scenario has been selected. The reason is (i) the difference in $D_F$ for GR1 and GR2 scenarios is larger compare to the other NBS scenarios, and (ii) GR1 is the only scenario that presents a different hydrological response (i.e., the highest percentage difference on peak flow was found for EV3).

Here, the GR1 scenario was taken as the reference scenario, we assume that the extremes of rainfall (hot spots) only falls on the green roofs of the GR1 scenario. With this respect, we binary re-distributed the rainfall in space during the 3 min that lasted at the largest rainfall peak of the EV3U, as illustrated on Figure 4c-f. Namely, the 'hot spots' of the EV4-EV6 synthetic rain events are strictly intersected with the GR1 distribution, while the GR2 scenario is not. Contrary to EV4-EV6, EV7 corresponds to the 'no rain' situation on GR1 during the same 3 min.

As shown in the hydrographs (Figure 18), the peak flow of GR1 scenario was expected to be less than that of GR2, and this is confirmed for EV4, EV5, and EV6. For EV4 and EV5, with the same maximum intensity of 55 mm $h^{-1}$, the hydrographs of these two events significantly differ, with the peak flow decreasing by a factor 2 for EV5. However, the only difference in the rainfall inputs is that there is zero rainfall outside of the GRs during the 3 min rainfall peak. The percentage difference on peak flow ($PD_{Qp}$) and total runoff volume ($PD_V$) of GR1 and GR2 scenario under the EV4 is around 5 %, and 4.3 %, respectively (see Figure 19). For EV5, the $PD_{Qp}$ and $PD_V$ increase to 20.7 % and 7.8 %, respectively. This confirms that without the impact of runoff that generated by other land uses, the intersection effects increase considerably with the high rainfall intensity, also increasing the NBS effectiveness. For the EV6, the maximum rainfall intensity during the 3 min has been decreased to 17 mm $h^{-1}$. This was sufficient to further reduce

the peak flow during the principal rainfall maximum. For this event, the $PD_{Op}$ and $PD_V$ values drop to 3.5 % and 1.8 %, respectively. This indicates that the intersection effects is less significant for the rainfall with the lower spatial variability of rainfall intensity. As expected in the EV7 scenario, because of zero rainfall intersected with the GRs in GR1 scenario, the peak flow of GR2 remains slightly smaller than that of the GR1, with the $PD_{Op}$ and $PD_V$ values of only 2.1 % and 1.4 %, respectively.

The results of this synthetic experiment firstly confirm that there is a complex interplay between the spatio-temporal intensity of precipitation and the runoff received from other parts of the watershed. Furthermore, this experiment strengthened the intersection effects on the GR scenarios. These intersection effects can be more significant for the rainfall events with higher spatial variability. Finally, as the rainfall fields are always variable in space and time, to make the most of the benefits of NBS for stormwater management, the results suggest to implement NBS scattered in the catchment, but with a higher fractal dimension $D_F$. This will combine a lower investment with the maximum return, preventing NBS from concentrated in certain specific places.

[Figure]

**Figure 4: (a) Time evolution of rainfall rate (mm h$^{-1}$) and cumulative rainfall (mm) of the EV3U over the whole catchment (the period between the red dash lines is the selected period for creating the EV4); (b) time evolution of catchment-averaged rainfall rate and cumulative rainfall of the EV4 over the whole catchment; (c) the rainfall intensity at the largest rainfall peak (distributed) over the Guyancourt catchment for the EV4 (the red pixels are the location of GRs in GR1 scenario with the highest rainfall intensity in space), the rainfall of other areas are uniform; (d-f) the rainfall intensity at the largest rainfall peak (distributed) over the Guyancourt catchment for the EV5, EV6 and EV7, respectively.**

[Figure]

**Figure 18. Simulated flow (m³ s⁻¹) of GR1 and GR2 scenario under the four syntactic rainfall events.**

[Figure]

**Figure 19. (a) Percentage difference on peak flow of GR scenarios under the four syntactic rainfall events; (b) percentage difference on total runoff volume of GR scenarios under the four syntactic rainfall events.**

*__Referee comments 2.4 Verification metrics:__*

*In the methods and results, the percent error is used. However, the authors compare different scenarios and approaches with each other. The percent error is generally used to compare a result with the true value or at least the theoretical value. It gives the feeling of an error, whereas you are not sure which one is right or wrong. Hence, using the percent difference throughout the manuscript may be a better choice.*

**__Author's answer:__**

Thank you for the suggestion, the authors are taking the percentage difference as the metrics throughout the manuscript.

The corresponding abbreviation for percentage difference on peak flow was modified to '$PD_{Qp}$', and the percentage

difference on total runoff volume was modified to '$PD_V$'.

***Referee comments 2.5 The results in a larger perspective***
*In the last section of the results and the conclusions, I miss a bit the significance of the results. How does this relate to other literature and are the intersection results of importance when compared to the large effect of uniform or non-uniform rainfall inputs?*

**Author's answer:**
Thanks to referee's constructive comments, the authors modified section 4.3 (in the revised manuscript is section 4.4), and added the following underlined sentences to enhance the comprehension and readability of the manuscript:

[revised manuscript text omitted]

*In addition, what can the authors say about the effects of the chosen events on the results? This is of course a case study, but I would encourage the authors to put the results a bit into perspective.*

**Author's answer:**

Thanks to referee's constructive comments, the authors modified the conclusion and strengthen the understanding of the effects of the chosen events.

**Conclusion:**

In our specific case, the GR scenarios are more sensitive to the spatial variability of rainfall and the spatial arrangement of GR measures, while the performances of RG scenarios and combined scenarios are more stable under any condition. Apparently, these findings already give some incites to decision-makers on *Why* they need to prioritize given NBS within the urban planning process.

Although the rainfall events selected for this study were not extreme events, they cover a rather broad spectrum of spatio-temporal variability in rainfall, and they are very typical precipitations in the Paris region. The simulation results can serve as a reference for future urban planning in this region. For example, the results of three different impacts (i.e., the spatial variability of precipitation, the spatial distribution of NBS, and the intersection effects) on the performance of NBS scenarios are useful for decision-makers, targeting for an actual project.

However, larger precipitation samples, including extreme rains, as well as NBS monitoring data will be helpful to get a better knowledge of somehow universal solutions and provide answers on *How* to prioritize these NBS. With respect to this perspective, the obtained results already demonstrated that new scale-independent indictors, like the fractal dimension $D_F$ applied in this study, will be essential for more profound quantitative evaluation of the diversity of combined impacts, including for other heterogeneous catchments. Therefore, this study have an important potential impact, due to its originality with respect to the nonlinear tools used to address such practical issues, and its relevance in interdisciplinary applications. This suggests to pursue the development of original tools to get new insights into the scaling complexity of flows in urban hydrology.

**3 Specific comments**

*Page 1, line 13: "of their hydrological responses sensitively depends". Do you mean that the hydrological model results depend on the mentioned two processes?*

**Author's answer:**

Thanks to the referee's suggestion, we rephrase this sentence (at line 13) as: 'However, the assessment of NBS performance still requires further modelling development because the hydrological modelling results strongly depends on the representation of multiscale space variability of both the rainfall and the NBS distribution.'

*Page 1, line 24: It is not more pronounced, but there is a somewhat significant effect for the two NBS scenarios the authors mention.*

**Author's answer:**

Thanks to the referee's suggestion, the sentence (at line 24, in the revised manuscript is line 26) was rephrased as 'Finally, the intersection of the spatial variability of rainfall and the spatial arrangement of NBS produces a somewhat significant effect on the peak flow of green roof scenarios and the total runoff volume of combined scenarios.'

*Page 1, lines 26 – 27: What kind of flooding do you mean (i.e., flash, coastal, urban, fluvial and/or pluvial floods)? In case of just urban flooding, the sentence is fine like this. Otherwise, there are more essential drivers, such as land subsidence in deltas, deforestation, etc. (add some extra references in that case). Nevertheless, I would suggest to change "The increasing of extreme flooding risks" into "The increase of extreme flood risk" (possibly with an indication of what kind of flood risk).*

**Author's answer:**

Thanks to the referee's suggestion, the sentence (at lines 26-27, in the revised manuscript is lines 29-30) was modified as: 'The increased risk of flooding from urban storms appears to be closely linked to two key factors: rapid urbanization and climate change (Lovejoy and Schertzer, 2013)'.

*Page 2, line 37: "parallel concepts". Perhaps I am not familiar enough with the concepts, but what kind of parallel concepts are meant here?*

**Author's answer:**

The 'parallel concepts' refers to some 'similar concepts' that are used in other countries or regions.

Thanks to the referee's suggestion, the sentence (at line 37, in the revised manuscript is lines 38-41) was modified to: 'To some extent, the NBS concept builds on and supports similar widely used concepts (Berry et al., 2015; Bozovic et al., 2017), like the Low Impact Development (LID), or Blue-Green Infrastructure (BGI), as well as some other more local ones, like the Water Sensitive Urban Design (WSUD) from Australia (Morison and Brown, 2011) or 'Sponge city' proposed recently in China (Chan et al., 2018).'

*Page 2, line 44 – 45: For completeness, also write why the fully-distributed models are rarely used.*

*Page 2, line 54: "Indeed, such models should better assess the hydrological performances of NBS on a smaller scale." Although I agree with the need for such models, be careful with this statement. It stands or falls with the presence of reliable high resolution model forcing and parameterization (among others).*

**Author's answer:**

Thanks to the referee's suggestions for the lines 44-45, and the line 54, the authors added the following underlined sentences (at lines 47-52 of the revised version) and removed the sentence at line 54 (previous version) to the revised version of the manuscript:

Indeed, as underlined by Fry and Maxwell, (2017), and Her et al. (2017), fully-distributed models are rarely used (Versini et al., 2016; Hu et al., 2017; Versini et al., 2018). While there is a general consensus that these models should better assess the hydrological performances of NBS implemented at smaller scales, the deployment of the fully distributed models has been stuck for some time by three main factors: *(i)* availability of reliable high resolution forcing, *(ii)* complex interactions between the processes, and *(iii)* reliable parameterisation process (e.g., Imhoff et al., 2020).

*Page 2, line 49: What do you mean by complicated urban catchments?*

**Author's answer:**
At the line 49 (in revised manuscript is line 54), the complicated urban catchments refer to the catchments with a very high heterogeneity.

Thanks to the referee's suggestion, this sentence was modified to 'Nevertheless, Rossman et al. (2010) demonstrated that SWMM has some serious limitations for reflecting the heterogeneity of urban watersheds, which in turn presents some difficulties to sustainably replicate hydrological responses to various urban land uses.'

*Page 2, line 55: "lack of high-resolution rainfall data". What is the desired resolution the authors are looking for? I.e., are rainfall estimates from X-band radars, commercial microwave links or personal weather stations a solution? I am aware of the challenge of (reliable, high-resolution) rainfall estimation in urban areas, but be a bit more specific about what the limitation is.*

*Page 2, lines 59 – 61: In addition, what about the time variability of rainfall. What can you say about the importance of that?*

**Author's answer:**

Thanks to the referee's suggestion, the authors added the following underlined sentences (at lines 61-70 of the revised version) to enhance the comprehension and readability of the manuscript:
At the same time, due to the long-standing challenge of the availability of reliable and high-resolution spatio-temporal precipitation measurements in urban areas, some studies have been devoted to assessing the performance of NBS under the simplifying assumption of an uniform rainfall, hence the impact of spatial rainfall variability in the heterogeneous urban context has not been considered (Holman-Dodds et al., 2003; Gilroy and McCuen, 2009; Qin et al., 2013; Versini et al., 2018; Zhu et al., 2019; Guo et al., 2019). A strong impact of the temporal variability of precipitation on the response of the watershed is generally well recognised (Schertzer et al., 2010; Ochoa-Rodriguez et al., 2015; Gires et al., 2015). Qin et al. (2013) also investigated the performance of some NBS, such as swales, porous pavements and green roofs, as a function of peak precipitation intensity. Whereas the temporal variability of precipitation, even intuitively, forces the dynamics of the retention capacity of the NBS, the impact of the spatial variability of precipitation in the heterogeneous urban context has not yet been studied in its full extent.

*Page 2, lines 59 – 61: This is the first time you introduce the intersection between the spatial variability of rainfall and the NBS. Although I do understand what you mean, I would suggest devoting one sentence here to explain what is meant by that.*

**Author's answer:**
Thanks to the referee's suggestion, the authors added the following underlined sentences (at lines 70-73 of the revised version) to enhance the comprehension and readability of the manuscript:

However, the hydrological responses of NBS (model outputs) can largely depend on: (i) the variability of the rainfall

fields, (ii) the spatial distribution of the NBS, and (iii) their intersection. Indeed, the rainfall and the NBS represent two heterogeneous fields that do not coincide, which implies that the overall performances of NBS scenarios simulated with uniform rainfall or lumped/semi-distributed model may not be entirely convincing.

*Page 3, line 68: "Two different types of rainfall data". The data source is the same, so I would rather suggest calling it two different rainfall processing approaches (gridded and catchment-averaged).*

**Author's answer:**
According to referee's suggestion, this sentence was modified to 'Two different rainfall processing approaches (gridded and catchment-averaged) of three typical rainfall events in the Paris area are used as meteorological inputs:'

*Page 4, lines 104 – 105: What is the rainfall amount that corresponds to this return period? That would make it easier to relate the studied events to the drainage capacity.*

**Author's answer:**
According to the referee's suggestion, the authors added the following underlined sentences (at lines 124-133 of revised version) to enhance the comprehension and readability of the manuscript:

The rainfall amount corresponding to the mentioned return periods (from 2 to 10 years) depends on the considered duration (usually equal to the concentration time). So this duration value depends on the location of pipes in the catchment and its upstream area. Here are the corresponding values for different durations that can be found on the studied watershed (by using the Montana coefficients):
- Duration 5 minutes: 187 mm/h for T = 10 years and 125 mm/h for T = 2 years
- Duration 30 minutes: 50 mm/h for T = 10 years and 31 mm/h for T = 2 years
- Duration 1 hour: 30 mm/h for T = 10 years and 18 mm/h for T = 2 years
- Duration 2 hours: 20mm/h for T=10 years and 13 mm/h for T=2 years
Regarding the properties of the three selected rainfall events (Table 1), the drainage system seems to have the capacity to drain the rainfall intensities on these durations. Nevertheless, we do not have any information about the exact duration range that was considered for the design (durations smaller than 30 minutes are usually not considered).

*Page 4, lines 113 – 115: Could you tell a little bit more about how this tool works?*

**Author's answer:**
According to referee's suggestion, the authors added the following underlined sentences (at lines 137-146 of revised version) to enhance the comprehension and readability of the manuscript.

As displays Figure 1, some vulnerable areas and buildings subject to a risk of waterlogging were defined in the Guyancourt catchment by using the ModelBuilder of ArcGIS software (a geoprocessing model, for identifying landscape sinks [https://learn.arcgis.com/en/]). This geoprocessing model is based on a sequential chain of GIS analysis tools, it firstly finds the landscape sinks on the DEM with the help of ArcGIS hydrology tools (https://desktop.arcgis.com).

We firstly identify the landscape sinks, on this figure, the blue spots represent the low-lying areas with a total area of 0.6 km$^2$ that can be easily flooded by stormwater (average rainfall depth of 53 mm). Then, the locations of the landscape sinks can be compared with the locations of existing buildings, and the buildings that are situated inside or adjacent to the landscape sinks are defined as the vulnerable buildings. Correspondingly, the yellow spots indicate these vulnerable buildings on the figure.

*Page 8, lines 245 – 251: This is a clear explanation, which would be even more valuable when placed at the start of the section.*

**Author's answer:**
Thanks to the referee's suggestion. We have moved this paragraph to the lines 298-306 of the revised manuscript.

*Page 11, lines 317 – 324: It would be good to place a reference to Nash & Sutcliffe (1970) here.*

**Author's answer:**
Thanks to the referee's suggestion. The reference was added in our revised manuscript (at line 414 of the revised manuscript) as: 'The Nash-Sutcliffe Efficiency (NSE ≤1) is an indicator generally used to verify the quality of the hydrological model simulation results, described as follows (Nash and Sutcliffe, 1970):'

*Page 11, lines 328 – 338: The authors already describe some model results here. This fits better in the results, e.g. as first subsection. Side note, the model indeed performs well for the given study area.*

*Page 11, lines 339 – 340: Although I do agree with this conclusion, can the authors say something about the model performance (regarding simulated fluxes and/or states) on the grid level or at the sub-catchment scale? On the used high spatial resolution and in an urban setting, I know this is challenging. However, it would further support your conclusions, especially because you are focusing on spatial variability in the results.*

**Author's answer:**
Thanks to your suggestion. We have moved the results of validation to the Section. 4.1 (revised version) and added some discussions in the revised version of the manuscript (at lines 428-449).

**4 Results and discussion**

**4.1 Validation of baseline scenario**
Regarding the water levels observed and those simulated in the baseline scenario, the model indeed performs well for the studied area. The *NSE* coefficients and the *PE* indicators validated Multi-Hydro's performance (see Table 4). Indeed, for the three distributed rainfall events (Figure 9), the *NSE* are larger than 0.9, and *PE* are lower than 5 %. For the uniform rainfall event of EV2, the model represents the water levels with *NSE* equal to 0.95, and *PE* equal to 1.96 %: only a slight overestimation of the observed water levels is observed between hours 4 to 7. For the uniform rainfall of EV1 and EV3, the temporal evolutions of simulated water levels slightly underestimate the observed ones, with *NSE* around 0.8, as well as *PE* around 7 %. Regarding the temporal evolutions of simulated water levels under

the distributed rainfall of EV1 and EV3, they are more consistent with the observed ones. The reason is that the rainfall intensities of the distributed rainfall are generally higher than those of the uniform rainfall at the storage basin location. Namely, in uniform rainfall events, the accumulated water levels in the storage basin are less than that of in distributed rainfall events. Overall, the distributed rainfall gives slightly better results, and the simulated water levels using uniform rainfall also match sufficiently well the observed ones to validate the Multi-Hydro implementation in the Guyancourt catchment.

Regarding the validation results, the scalability of Multi-Hydro allowed us to define the optimal resolution to finely reproduce the spatial heterogeneity of the watershed. Remember that this resolution is the ratio between the external scale of the watershed and the scale of the grid. The heterogeneity mentioned above propagates from the smallest scale to the largest, impacting the simulation results in any through the hierarchy of spatial scales of the watershed. It should be understood that the selected 10 m grid scale is not the smallest scale possible, but the optimal one to ensure a good balance between, for example, sufficient heterogeneity and the required quantity of the data required, again in precision valuable and involved computing time involved. As discussed in Section 3.2, the spatial heterogeneity for each of the NBS scenarios evolves with the fractal dimension on two scale ranges: the asset implementation scales (10m - 80m) and the larger basin scales. Such an evolution remains fully compatible with the intrinsic scalability of Multi-Hydro, which makes it particularly suitable and sufficiently reliable to study the impacts of the spatial variability of hydrological responses in different NBS scenarios.

*Page 12, lines 361 – 363: This is exactly what you expect for the baseline situation. The difference in total runoff volume should not be too different, because the total rainfall volume should be the same for the gridded and uniform rainfall inputs. The small differences are an effect of differences on the grid scale (storage capacity, evapotranspiration, etc.), which are differently modelled when the input is uniform or non-uniform.*

**Author's answer:**

Thanks to your suggestion, we modified this sentence (lines 470-474 of revised manuscript) as follows:

For the baseline scenario, it is noticed that the $PD_{Op}$ is more pronounced than $PD_V$ for all rainfall events. These results can be explained by the fact that the spatial variability of rainfall intensity at the largest rainfall peak is strong in all three rainfall events, while the total rainfall volume for the distributed and uniform rainfall inputs is the same. This small $PD_V$ is influenced by the differences on the grid scale (storage capacity, interception, etc.), which are differently modelled when the input is uniform or non-uniform.

*Page 14, lines 433 – 437: I think this is one of the most interesting (and important) results of the study. However, it is not very easy for the reader to make the comparison based on the figures (e.g. figures 14 and 16). Could the authors add a subplot to figure16 indicating the differences between the two scenarios (so the difference between uniform and non-uniform rainfall plus the difference between the scenarios) and tell somewhat more about it?*

**Author's answer:**

As referee suggested, we add a subplot to Figure 16 (in the revised manuscript is Figure 17). The corresponding figures and discussions can be found in the answer for *2.5 The results in a larger perspective.*

*Page 15, conclusions: The abbreviations used throughout the text, are also directly used in the conclusions. If you read the entire text, this is clear, but for readers who quickly skim through the abstract and conclusions, I would suggest to (re-)introduce the meaning of these abbreviations.*

**Author's answer:**

Thanks to the referee's suggestion. These abbreviations are re-introduced in the revised manuscript.

*Page 16, lines 479 – 480: "However, the RG scenarios appear to be less affected by the intersection effects, with a difference lower than 3% on peak flow and lower than 1 % on total runoff volume." This is indeed supported by the results, but in the results, you also discuss the reason for this small effect on the peak flow and runoff volume. It would be good to include that here too.*

**Author's answer:**

Thanks to the referee's suggestion. This sentence has been modified in the revised version of manuscript: "However, the RG scenarios appear to be less affected by the intersection effects, with a difference lower than 3 % on peak flow and lower than 1 % on total runoff volume, mainly due to their high storage capacity."

*Page 16, lines 481 – 485: I fully agree with the authors that this hints towards using fully distributed hydrological models over semi-distributed or lumped models, but that is not exactly what is shown in the results. The authors do not benchmark the result with semi-distributed or lumped models, but rather focus on the rainfall variability and NBS variability on the discharge response. I would like to ask the authors to rephrase this paragraph a bit.*

**Author's answer:**

Thanks to the referee's suggestion. Lines 481-485 (in the revised manuscript is 643-647) have been modified to:

'The study of hydrological response in various NBS scenarios resulting from the multi-scale spatial variability of precipitation and the heterogeneous distribution of NBS hints towards using fully distributed hydrological models over semi-distributed or lumped models. Indeed, the fully distributed model has been shown to be able to take into account these small-scale heterogeneities and propagate their effects to watershed scales, while parameterizing or smoothing out some critical heterogeneity, as done in non-fully distributed models, may bias its predictions.'

*Figures overall – Make sure the font size is readable and approximately the same font size is used for all the figures in the manuscript.*

**Author's answer:**

Thanks to the referee's suggestion, we updated the figures with the same font size in our revised manuscript.

**_Referee comments 4 2 Technical corrections_**
*Page 1, lines 17 – 19: For readability, I would suggest making two separate sentences out of this one.*

As referee suggested, this sentence (at lines 17-19, in the revised manuscript is lines 20-21) was modified in two separate sentences.

Then, a fully-distributed and physically-based hydrological model (Multi-Hydro) was applied to consider the studied catchment and these NBS scenarios with a spatial resolution of 10 m. Two approaches for processing the rainfall data were considered for three rainfall events: gridded and catchment-averaged.

*Page 1, line 21: ", which is more pronounced than those of the total runoff volume." Do you mean, ", which is a stronger effect than the effect on the total runoff volume."?*

**Author's answer:**

As referee suggested, this sentence (at line 21, in the revised manuscript is line 22) was revised to: These simulations show that the impact of spatial variability of rainfall on the uncertainty of peak flow of NBS scenarios ranges from about 8 % to 18 %, which is more significant than those of the total runoff volume.

*Page 2, line 30: "results in rainfall transfer into runoff rapidly" becomes "result in a rapid transfer of rainfall into runoff".*

**Author's answer:**

Thanks to your suggestions. The sentence (line 30, in the revised manuscript is line 32) was revised to: Impervious surfaces directly connected to grey infrastructures result in a rapid transfer of rainfall into runoff.

*Page 2, line 31: "The approach of expanded and upgraded the capacity of the existing drainage system" becomes "Expanding and upgrading the capacity of the existing drainage system".*

**Author's answer:**

Thanks to your suggestions. The sentence (line 31, in the revised manuscript is line 33) was revised to: Expanding and upgrading the capacity of existing drainage systems has proven to be costly and unsustainable, which is challenging to realize in highly urbanized cites (Qin et al., 2013).

*Page 3, line 62: "such mentioned" becomes "the mentioned" and "over higher spatial resolutions" becomes "for higher spatial model resolutions".*

**Author's answer:**

Thanks to your suggestions. The sentence (line 62, in the revised manuscript is lines 73-74) was revised to: Therefore, the mentioned impacts remain to be investigated, especially for higher spatial resolutions, by using the space-time rainfall fields together with a fully-distributed model, allowing for heterogeneous NBS scenarios.

*Page 4, line 109: "a clear tendency towards growing number of somewhat shorter, but much heavier rainfall events, was perceived for this region" Suggested change: "A clear tendency towards a growing number of shorter duration, but higher intensity rainfall events is perceived for this region".*

**Author's answer:**

Thanks to your suggestions. The sentence (line 109, in the revised manuscript is lines 134-136) was revised to: A clear tendency towards a growing number of shorter duration, but higher intensity rainfall events is perceived for this region (Hoang et al., 2010), causing in recent years a large amount of fast surface runoff and higher peak flow rates.

*Page 6, line 184: Remove the ')' before the end of the sentence.*

**Author's answer:**

Thank you for your carefully reading. The ')' (at line 184) was removed in the revised manuscript.

*Page 7, line 201: "simulated under both different types of rainfall" becomes "simulated for both rainfall inputs".*

**Author's answer:**

Thank you for your suggestion, the sentence (line 201, in the revised manuscript is line 278) was revised to: For achieving the purpose of the study, a series of NBS scenarios were created and simulated for both rainfall inputs (described in Sect. 2.2).

*Page 18 and 19, lines 559 – 567: The references are not alphabetical here. Perhaps also at other lines.*

**Author's answer:**

Thanks to the referee's suggestion. The reference order was modified in our revised manuscript.

*Figure 2: I may be wrong, but it seems that the legend colour for forest is not exactly the same as the colour used in the map. Figure 3: The rainbow colour map is not always intuitive, also here with respect to the rainfall amounts. I recommend using a more intuitive colour map. Some explanations and inspiration can be found in Crameri et al. (2020).*

**Author's answer:**

Thanks to your suggestion and reference. The legend colour for the forest was changed in the map (Figure 2). The rainbow colour map was changed for Figure 3

[Figure]

**Figure 2: Left: land use map (baseline scenario); right: drainage system with four conduits (4541, 4542, 4543, and 4544) highlighted.**

[Figure]

**Figure 3: Top: The rainfall intensity at the largest rainfall peak (per radar pixel) over the Guyancourt catchment area for the three studied rainfall events; middle: cumulative rainfall depths (per radar pixel) over the Guyancourt catchment area for the three studied rainfall events; bottom: time evolution of rainfall rate (mm h⁻¹) and cumulative rainfall (mm) of the three uniform rainfall events over the whole catchment.**

*Figure 10: The authors refer in the caption to (a) EV1, (b) EV2, (c) EV3, but the letters (a) – (c) are not shown in the figure.*

**Author's answer:**
Thanks to your suggestion. The letters (a) – (c) were added in the revised figure (in the revised manuscript is Figure. 11).

[Figure]

**Figure 10: Simulated flow (m³ s⁻¹) of the baseline scenario under three distributed rainfall events and three uniform rainfall events: (a) EV1, (b) EV2, (c) EV3.**

*Figure 12c: It would be better to show the ratio on a logarithmic axis.*

**Author's answer:**
Thanks to your suggestion. The ratio of peak flow between the scenarios under the distributed rainfall and the scenarios under the uniform rainfall was plotted on a logarithmic axis in the revised manuscript.

[Figure]

**Figure 12. (a) The ratio of peak flow between the scenarios under the distributed rainfall and the scenarios under the uniform rainfall; (b) percentage _difference_ on peak flow of the baseline scenario and the first set of NBS scenarios under the three distributed rainfall events and the three uniform rainfall events; (c) percentage _difference_ on total runoff volume of the baseline scenario and the first set of NBS scenarios under the three distributed rainfall events and the three uniform rainfall events.**

_Figure 14: I spotted a minor typo in the figure title (Percenatge instead of percentage)._

**Author's answer:**
Thanks to the referee's suggestion. The misspelling was revised for the corresponding figure.

_5 References_

_Crameri, F., Shephard, G.E. Heron, P.J (2020). The misuse of colour in science communication. Nature Communications, 11, 5444 (2020). https://doi.org/10.1038/ s41467-020-19160-7._

[revised manuscript text omitted]